# Permafrost is warming at a global scale

Boris K. Biskaborn ⬡ et al.#

Permafrost warming has the potential to amplify global climate change, because when frozen sediments thaw it unlocks soil organic carbon. Yet to date, no globally consistent assessment of permafrost temperature change has been compiled. Here we use a global data set of permafrost temperature time series from the Global Terrestrial Network for Permafrost to evaluate temperature change across permafrost regions for the period since the International Polar Year (2007–2009). During the reference decade between 2007 and 2016, ground temperature near the depth of zero annual amplitude in the continuous permafrost zone increased by 0.39 ± 0.15 °C. Over the same period, discontinuous permafrost warmed by 0.20 ± 0.10 °C. Permafrost in mountains warmed by 0.19 ± 0.05 °C and in Antarctica by 0.37 ± 0.10 °C. Globally, permafrost temperature increased by 0.29 ± 0.12 °C. The observed trend follows the Arctic amplification of air temperature increase in the Northern Hemisphere. In the discontinuous zone, however, ground warming occurred due to increased snow thickness while air temperature remained statistically unchanged.

. Correspondence and requests for materials should be addressed to B.K.B. (email: boris.biskaborn@awi.de). #A full list of authors and their affiliations appears at the end of the paper.

O ne quarter of the Northern Hemisphere and 17% of the Earth's exposed land surface is underlain by permafrost[1], that is ground with a temperature remaining at or below 0 °C for at least two consecutive years. The thermal state of permafrost is sensitive to changing climatic conditions and in particular to rising air temperatures and changing snow regimes[2–7]. This is important, because over the past few decades, the atmosphere in polar and high elevation regions has warmed faster than elsewhere[8]. Even if global air temperature increased by no more than 2 °C by 2100, permafrost may still degrade over a significant area[9]. Such a change would have serious consequences for ecosystems, hydrological systems, and infrastructure integrity[10–12]. Carbon release resulting from permafrost degradation will potentially impact the Earth's climate system because large amounts of carbon previously locked in frozen organic matter will decompose into carbon dioxide and methane[13–15]. This process is expected to augment global warming by 0.13–0.27 °C by 2100 and by up to 0.42 °C by 2300[15]. Despite this, permafrost change is not yet adequately represented in most of the Earth System Models[14] that are used for the IPCC projections for decision makers. One major reason for this was the absence of a standardized global data set of permafrost temperature observations for model validation.

Prior to the International Polar Year (IPY, 2007–2009), ground temperatures were measured in boreholes scattered across permafrost regions. However, a globally organized permafrost data network and a standard reference period against which temperature change could be measured did not exist. One key outcome of the IPY was strenghtening the Global Terrestrial Network for Permafrost (GTN-P)[16,4]. This initiative established a temperature reference baseline for permafrost and led to an increase in the number of accessible boreholes used for temperature monitoring.

To analyze the thermal change of permafrost we assembled a global permafrost-temperature data set that includes time series of data attributed to the IPY reference boreholes. We compiled a time series for the decade from 2007 to 2016 that comprises mean annual ground temperatures $\bar{T}$, determined from temperatures measured in boreholes within the continuous and discontinuous permafrost zones in the Arctic (including the Subarctic), Antarctica and at high elevations outside the polar regions. The measurements were made at, or as close as possible to the depth of zero annual amplitude $Z^*$, where seasonal changes in ground temperature are negligible (≤0.1 °C). Rates of permafrost temperature change calculated for the 2007–2016 decade were indexed in each borehole to suppress near-surface and deep geothermal changes. Regional and global change rates were calculated as area-weighted means. To compare single borehole sites, due to the higher availability of full-year records after 2007, we ranked the temperature difference between the biennial means of 2008–2009 and 2015–2016. We used linear regression on $\bar{T}$ between 2007–2016 to estimate decadal change rates. To calculate annual departures, we compared consecutive years to the reference mean of 2008–2009. We concluded, that ground temperature near the depth of zero annual amplitude increased in all permafrost zones on Earth, that is continuous and discontinuous permafrost in the Northern Hemisphere, as well as permafrost in the mountains and in Antarctica. The observed trend followed increased air temperature and snow thickness, each in varying degrees depending on the region.

## Results

**Permafrost temperature changes.** Measurements from borehole sites established prior to the IPY generally indicated warming driven by higher air temperatures (Fig. 1)[4,17,18]. Our new data set

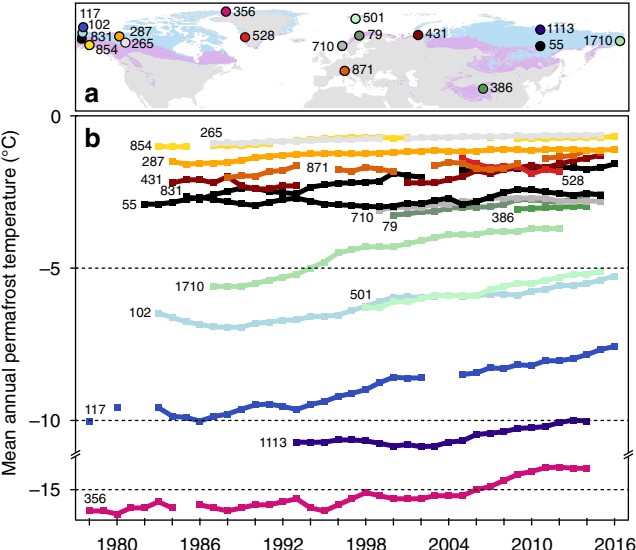

**Fig. 1** Long permafrost temperature records for selected sites. **a** Location of boreholes with long time-series data. Because some regions lack long temperature records, shorter temperature records from Greenland and Chinese mountains are included for comparison. Depth of measurements is according to the Global Terrestrial Network for Permafrost ID[16]: 24.4 m (ID 356), 20 m (ID 55, 79, 102, 117, 501, 710, 831, 1113, and 1710), 18 m (ID 386), 16.75 m (ID 871), 15 m (ID 854), 12 m (ID 287), 10 m (ID 265, 431), and 5 m (ID 528). The light blue area represents the continuous permafrost zone (>90% coverage) and the light purple area represents the discontinuous permafrost zones (<90% coverage). **b** Mean annual ground temperature over time. Colors indicate the location of the boreholes in a. Permafrost zones are derived from the International Permafrost Association (IPA) map[46]. World Borders data are derived from http://thematicmapping.org/downloads/world_borders.php and licensed under CC BY-SA 3.0 (https://creativecommons.org/licenses/by-sa/3.0/)

contains 154 boreholes of which 123 allow calculation of decadal temperature change rates based on adequate time series. The remaining 31 boreholes provide additional information on annual departures. Our results show that in the decade after the IPY permafrost warmed within 71 boreholes, cooled in 12, and remained unchanged (within measurement accuracy) in the remaining 40 (Fig. 2). The ground temperature rose above 0 °C in five boreholes, indicating thawing at the measurement depth of 10 m at $Z^*$. The largest increase of $\bar{T}$ over the observed reference decade between 2007 and 2016 was $0.39 \pm 0.15$ °C $dec^{-1}_{\text{Ref}}$ in the Arctic continuous permafrost zone. The greatest permafrost temperature changes observed in individual boreholes ($\Delta \bar{T}_b$) since 2008–2009 were 0.93 and 0.90 °C in northwestern Siberia (Marre Sale, 10 m) and northeastern Siberia (Samoylov Island, 20.75 m), respectively. The discontinuous permafrost zone experienced warming of $0.20 \pm 0.10$ °C $dec^{-1}_{\text{Ref}}$. The largest $\Delta \bar{T}_b$ since 2008–2009 of 0.95 °C was observed in southeastern Siberia, Magadan (Olsky pass, 10 m). Permafrost at this site started thawing after the IPY period at the measurement depth.

Mountain permafrost in the data set is mainly represented by boreholes in the European Alps, the Nordic countries, and central Asia. Although absolute $\bar{T}$ values in mountain permafrost are highly heterogeneous, depending on elevation, local topography, snow regime, and subsurface characteristics, changes in mountain permafrost temperatures were analyzed for all regions and settings[19] as one group. They can vary considerably, however, between sites of low and high ground ice content at temperatures just below 0 °C.

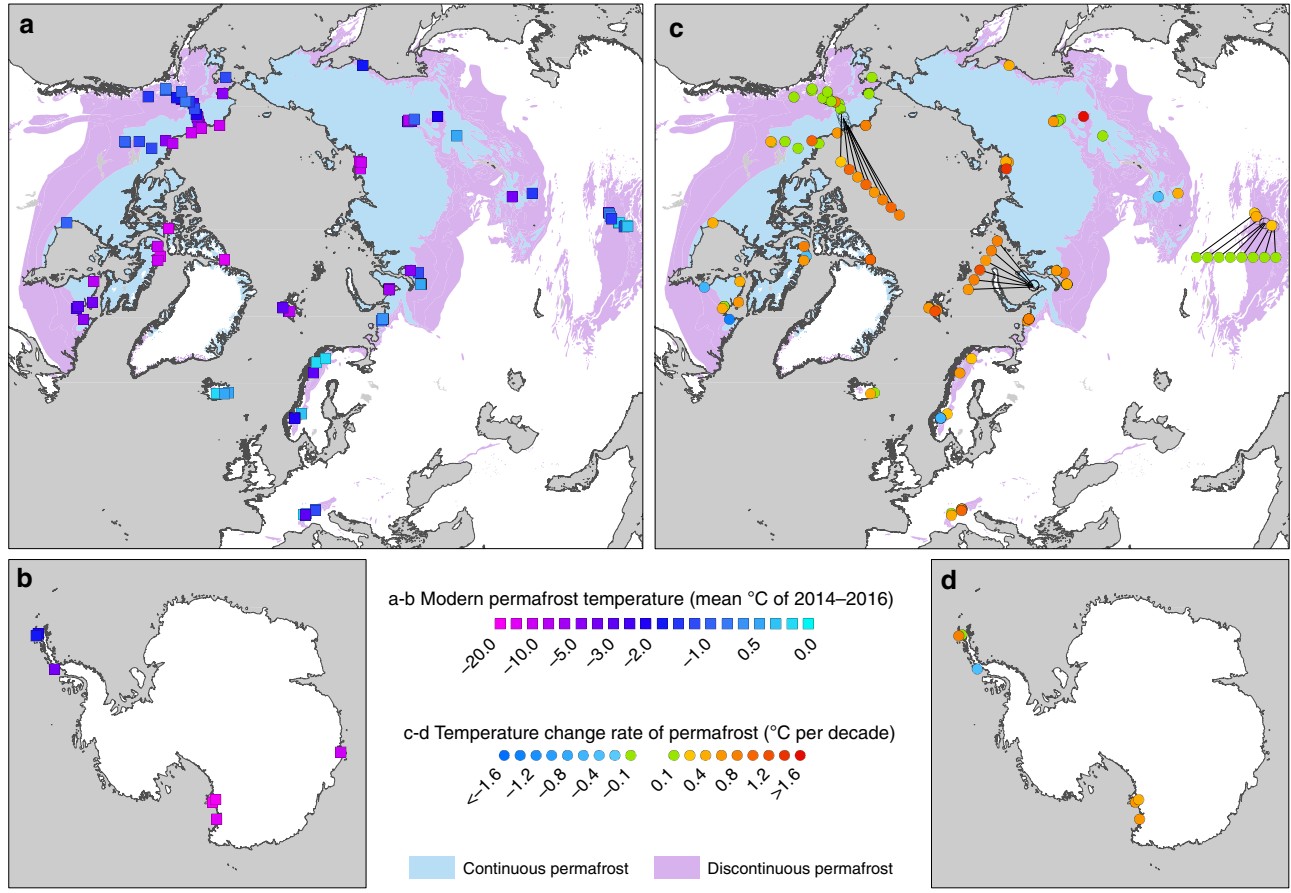

**Fig. 2** Permafrost temperature and rate of change near the depth of zero annual amplitude. **a, b** Mean annual ground temperatures for 2014–2016 in the Northern Hemisphere and Antarctica, $n = 129$ boreholes. **c, d** Decadal change rate of permafrost temperature from 2007 to 2016, $n = 123$ boreholes (Eq. 3). Changes within the average measurement accuracy of ~±0.1 °C are coded in green. Continuous permafrost zone (>90% coverage); discontinuous permafrost zones (<90% coverage). Permafrost zones are derived from the International Permafrost Association (IPA) map[46]. World Borders data are derived from http://thematicmapping.org/downloads/world_borders.php and licensed under CC BY-SA 3.0 (https://creativecommons.org/licenses/by-sa/3.0/)

Mountain permafrost $\bar{T}$ increased[20,21] by 0.19 ± 0.05 °C $dec_{\mathrm{Ref}}^{-1}$. The greatest $\Delta \bar{T}_b$ since 2008–2009 was 1.15 °C, observed in the Aldan mountain tundra of southern Yakutia, Siberia (Taezhnoe, 25 m).

On average, permafrost across zones warmed by 0.33 ± 0.16 °C over the reference decade in northern Asia and by 0.23 ± 0.11 °C $dec_{\mathrm{Ref}}^{-1}$ in North America. This difference is most likely due to stronger warming of the atmosphere over North Asia compared to North America, as indicated by reconstructed decadal air temperature changes (1998–2012) that showed cooling in Alaska[22].

Similar to warming of the Arctic continuous permafrost zone, the Antarctic permafrost warmed by 0.37 ± 0.10 °C $dec_{\mathrm{Ref}}^{-1}$. However, the remoteness of the continent and its limited accessibility resulted in far fewer boreholes drilled to $Z^*$ compared to the Northern Hemisphere. Consequently, permafrost temperature departures and trends were statistically not significant and had large uncertainty bands (Fig. 3d).

**Air temperature changes**. The relation between air and soil temperature development in permafrost regions is not straightforward due to highly variable buffer layers such as vegetation, active layer soils, or snow cover. To compare permafrost temperature changes to those in the atmosphere, we applied the same calculation method for each borehole site using mean annual air temperatures

($\hat{T}$) at 2-m height above ground level (Fig. 4a, d), spatially interpolated from the ERA Interim gridded reanalysis data set[23]. We calculated general snow thickness changes for Arctic sites in Fig. 4a, b. However, there is not, as yet, a reliable consistent data set on snow thickness applicable for high elevation regions or Antarctica.

The propagation of temperature change in the atmosphere downward to the depth of $Z^*$ can take up to several years, but the time varies depending on the surface characteristics, the subsurface ice content, and the soil thermal diffusivitiy[24,25]. We took this lag into account by averaging over the previous 4 years for each year considered, but there was no significant correlation at an annual resolution between permafrost temperature departures at $Z^*$ depth and 2-m air temperature anomalies derived from ERA Interim data alone (Fig. 4). This lack of correlation can be attributed to the discrepancy between the scale at which borehole observations are conducted and the spatial resolution of 80 km for the gridded air-temperature reanalysis data[26] and because in permafrost regions, the reanalysis output is more dependent on the model structure and data assimilation methods than in data-rich regions[27]; local micro- and secondary climate effects[28]; and buffering layers at the air-ground interface[5] that influence the thermal response of permafrost to short-term changes in air temperature.

Previous studies have shown that these surface effects, along with the thermal diffusivity of the underlying materials, act as a buffer that reduces the effect of short-term climate

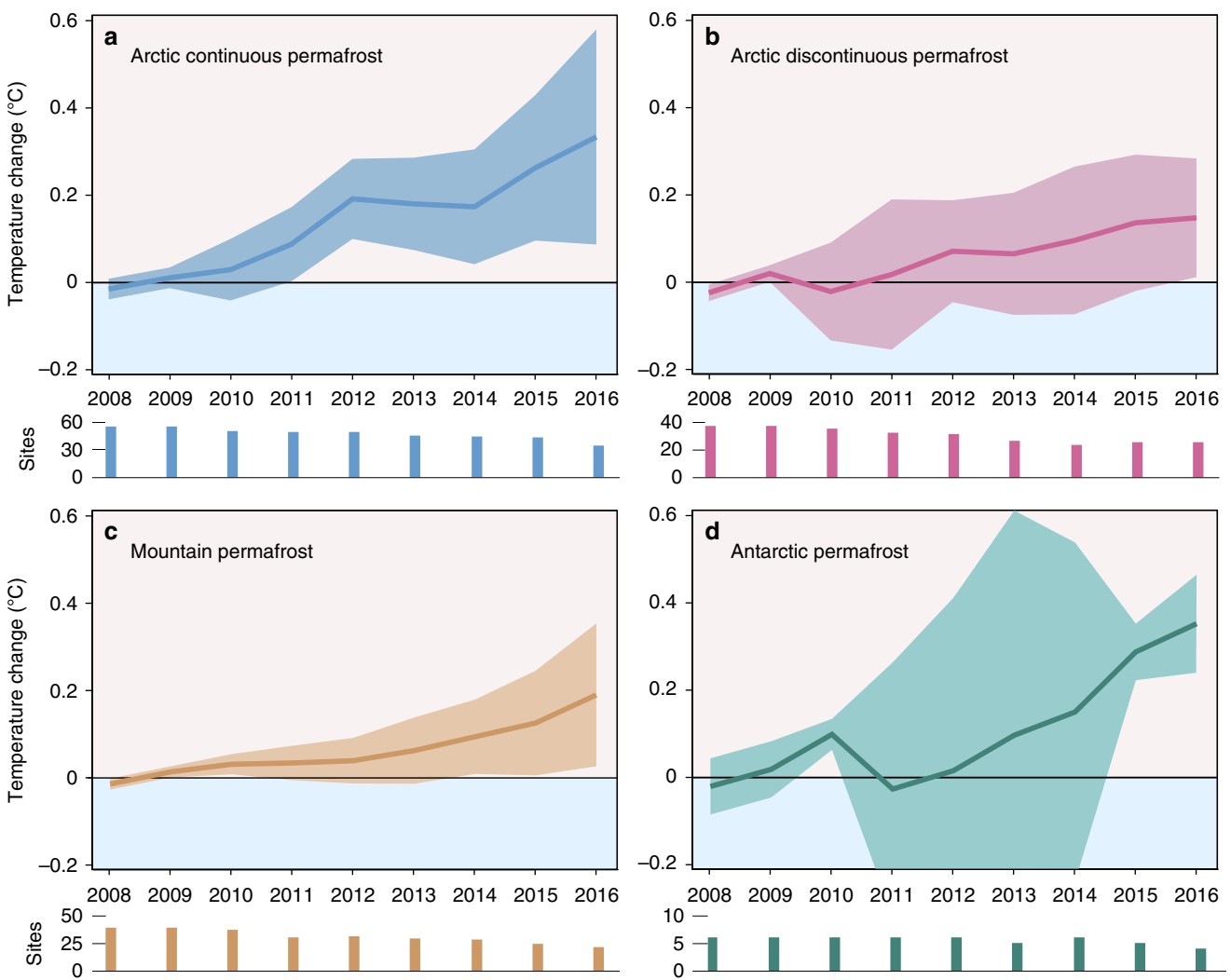

**Fig. 3** Annual permafrost temperature change. **a**–**d** Permafrost temperature departure $\Delta\bar{T}_{y,b}$ calculated from mean annual ground temperatures in boreholes near the depth of zero annual amplitude $Z^*$ relative to the 2008–2009 reference period. Mean values calculated as de-clustered, indexed area-weighted averages (Eq. 1). Temperature uncertainties are expressed at 95% confidence. Sample size shown is the number of borehole sites per year and region

variation[2,3,5–7,29]. Thus, short-term meteorological phenomena are increasingly attenuated and delayed with depth, and the mean permafrost temperature changes near the depth of $Z^*$ generally follow the atmosphere's long-term trend. Mean surface air temperature changes calculated from ERA Interim data at the borehole locations (Fig. 5b) are similar to those for permafrost temperature with respect to direction and order of magnitude. The decadal change rates of air temperature were estimated to $0.86 \pm 0.84\,°C$ per reference decade in the Arctic continuous permafrost zone, $0.63 \pm 0.91\,°C\ dec_{Ref}^{-1}$ in the Arctic discontinuous permafrost zone, and $0.1 \pm 0.50\,°C\ dec_{Ref}^{-1}$ in mountain permafrost. Air temperature trends in Antarctica (annual mean $0.10 \pm 0.55\,°C\ dec_{Ref}^{-1}$, June–August mean $-0.48 \pm 0.91\,°C\ dec_{Ref}^{-1}$, unweighted median $-0.12$, Fig. 5b), however, do not match the observed strong permafrost warming. This discrepancy is due to large climatic differences between the Antarctic Peninsula and eastern Antarctica[30,31], the small number of boreholes that fulfill the quality criteria, and the principal climate model bias in Antarctica[32].

Air temperature trends in the Arctic continuous permafrost zone correspond well with permafrost temperature change rates (Figs. 3a and 4a), suggesting that enhanced warming of permafrost in the High Arctic reflects the polar amplification of recent atmospheric warming[22]. However, in the Arctic discontinuous permafrost zone, air temperatures were statistically unchanged between 2006 and 2014 while permafrost temperatures increased. We found that snow dynamics, the time lag between air and ground temperature, and the latent heat effect serve as concurrent explanations for this phenomenon.

**Snow thickness changes**. The snow cover reduces the upward transfer of energy from the ground to the air during winter[33,34]. Distinct peaks in the mean snow depth in 2009, 2011 and from 2013 onward (Fig. 4a, b) suggest that the observed continued warming of discontinuous permafrost is facilitated by increasing snow thickness. Compared to the Arctic continuous permafrost zone, the mean snow cover in the discontinuous zone arrived about 1 week later, reached its maximum insulation 1 month earlier, and also disappeared half a month earlier. Compared to 2007–2009 the snow cover in 2014–2016 in the discontinuous zone started to form 13.7 days earlier, reached its maximum insulation effect 37.7 days earlier, and disappeared 9.3 days earlier (Fig. 4f). It was shown previously that a difference of only 10 days caused significant warming in Alaska[35]. Increases of shrub height and density that trap wind drifting snow is likely also a

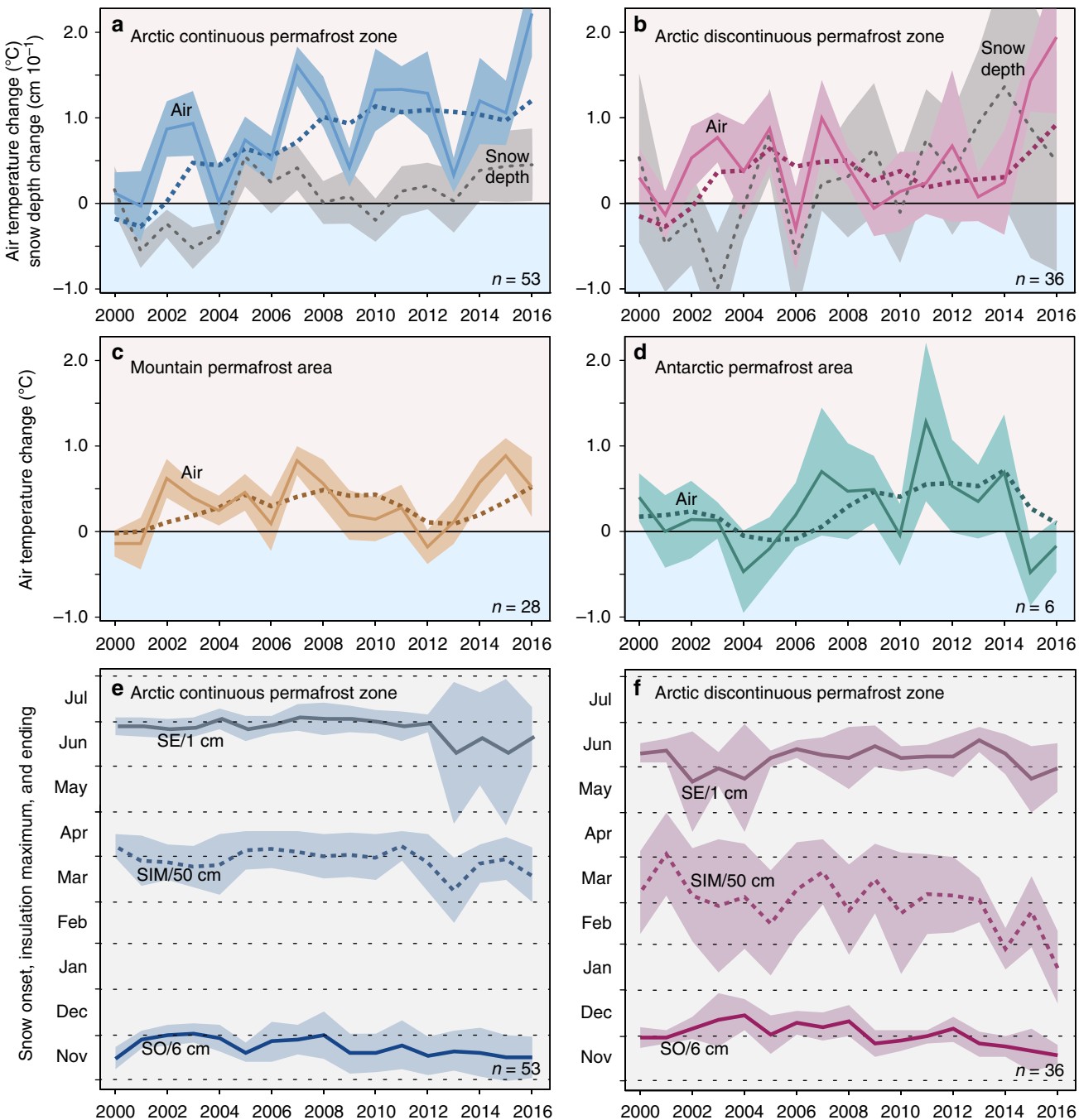

**Fig. 4** Annual air temperature and snow depth changes. **a–d** Air temperature anomaly $\Delta\hat{T}_{y,b}$ relative to the 1981–2010 reference period calculated from mean annual air temperatures at 2-m height above the ground level interpolated from the ERA Interim reanalysis data set. Mean values calculated as de-clustered, indexed area-weighted averages (Eq. 4). Dark colored dashed lines indicate 4-year end-point running means. Snow depth changes $\Delta\hat{S}_{y,b}$ in **a** and **b**, indicated in gray, calculated as the difference relative to the 1999–2010 reference period from the CMC reanalysis data set (eq. 5). **e, f** Onset of snow SO, snow insulation maximum SIM (dashed line), and the end of snow melt SE. Uncertainties are expressed as shading at 95% confidence. Sample size $n$ indicates the number of boreholes

contributing factor[36]. All of these changes provide evidence of increased protection of the ground from low temperatures during winter[37,38]. Snow timing differences within the continuous zone are less distinct but show a generally similar trend (Fig. 4e, f).

## Discussion

An important factor that explains the general discrepancy between mean annual temperature changes at $Z^\star$ in permafrost and the atmosphere is that permafrost progressively with depth "remembers" the surface temperature history of the past several years[25,39]. The temporal dimension of episodes with lower air temperatures between 2009 and 2013 in the Arctic (Fig. 4a, b), and around 2012 in the mountains (Fig. 4c), relative to preceding period of higher air temperatures, however, was not large enough to sustainably impact the general warming trend of permafrost.

We partly attribute the difference in ground temperature change between the continuous permafrost and the discontinuous permafrost zones to the latent heat effect. In this process, the ice-water phase change associated with warmer permafrost in the

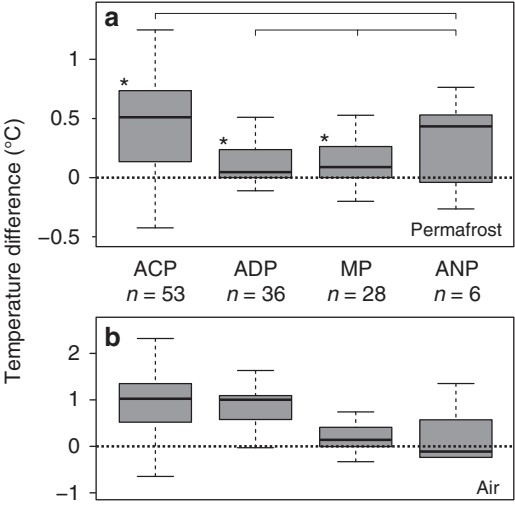

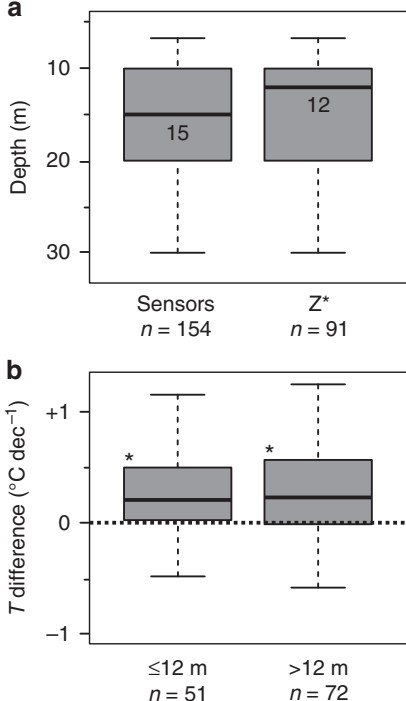

**Fig. 5** Decadal temperature change rates at permafrost borehole sites. **a** Boxplots showing the regional (unweighted) distribution of permafrost temperature change rates near the depth of zero annual amplitude $Z^*$ calculated for 2007–2016 in °C per decade (Eq. 3). * indicate significant difference to 0, $p < 0.05$ defined by the Wilcoxon Signed-Rank test. $p$ values ACP: 0.000, ADP: 0.002, MP: 0.016, and ANP: 0.156 (rounded to three digits). The Kruskal–Wallis test indicated $p$ values > 0.05 for couples that are tied with brackets in the graph. **b** Air temperature change rates at 2 m height above ground at borehole sites in °C per decade, calculated from the ERA Interim reanalysis data for 2004–2016, separated by regions. Symbols: $n$ number of boreholes, ACP Arctic continuous permafrost, ADP Arctic discontinuous permafrost, MP mountain permafrost, ANP Antarctic permafrost. Boxes represent 25–75% quartiles and whiskers are 1.5 interquartile ranges from the median. Medians are shown as black lines

**Fig. 6** Depth distribution of borehole temperatures. **a** Boxplots showing the depth distribution of temperature measurements (sensor depths) and of the zero annual amplitude $Z^*$ in the boreholes. **b** Temperature distribution in borehole sensors that are shallower (≤12 m) and deeper (>12 m) than the median of $Z^*$. * indicate significant difference to 0, $p < 0.05$ defined by the Wilcoxon Signed-Rank test; $p$ values ≤12 m: 0.000, >12 m: 0.000 (rounded to three digits). The Kruskal–Wallis test between these zones (**b**) resulted in $p = 0.908$, indicating that the zones are not significantly different to each other. Boxplots represent 25–75% quartiles and whiskers are 1.5 interquartile ranges from the median. Medians are shown as black lines and labeled with values. The number of boreholes (sensors), and the number of available $Z^*$ values is indicated by $n$

discontinuous zone (Fig. 2a, b) reduces the response of ground temperature to changes in air temperature[4]. Cold permafrost therefore exhibits a greater response to changing air temperature compared to permafrost with a temperature close to 0 °C[4,40].

The warming of permafrost observed since IPY continues the trends documented prior to IPY[41]. Our global analysis suggests that the future increases in air temperature projected under current climate scenarios[42] will result in continued permafrost warming. The duration of our time-series, however, does not yet permit predictive analysis of non-linear climate-permafrost relations as the latent heat effect is stronger near 0 °C and surface characteristics are not constant. However, observations of thaw at some of the observation sites demonstrate that the latent heat requirement cannot indefinitely delay permafrost warming down to depths of about 15 m observed in this study (Fig. 6), nor prevent the eventual thawing of permafrost. This could have wide implications in terms of permafrost degradation and release of greenhouse gases from decomposition of organic matter.

The SWIPA 2017 report[41] gave an estimate of 0.5 °C warming of permafrost in very cold areas such as the High Arctic since IPY (2007–2009). This is similar to our network observations of strong warming within the Arctic continuous permafrost zone and of continued warming elsewhere. The assessment of permafrost temperature trends presented in this paper can facilitate validation of models to project thawing of permafrost down to the depth of $Z^*$ and associated impacts with respect to feedbacks to the climate system.

The current global coverage of permafrost temperature monitoring is not yet ideal, due to the limited sampling in regions such as Siberia, central Canada, Antarctica, and the Himalayan and Andes mountains. Furthermore, even though the data used were quality checked and are as complete as possible, logistical challenges during fieldwork caused gaps in the time series. Better assessments of the evolution of the thermal state of permafrost, including consideration of non-linear system behavior, will benefit from ongoing efforts to enhance the global network spatially and extend the length of the record. Enhancing existing monitoring sites through co-location with meteorological stations could further improve understanding of microclimate and buffer-layer influences, and would also provide the data necessary for a comprehensive assessment of permafrost responses to ongoing climate change.

The newly compiled GTN-P data set has facilitated assessment of trends in permafrost temperatures and can also contribute to improved representation of permafrost dynamics in climate models and the reduction of uncertainty in the prediction of future conditions.

## Methods
**Field observations of permafrost temperatures**. Boreholes were established and temperatures were recorded during annually repeated fieldwork campaigns in polar and high-elevation areas. Temperature was measured either by lowering a calibrated thermistor into a borehole, or recorded using permanently installed multi-sensor cables[43]. Measurements were recorded either manually with a portable temperature system or by automated continuous data logging. At some borehole sites, permafrost thawed at the measurement depth during the period of observations. The criterion to include non-permafrost sites in the global change calculation was that ground temperatures near the depth of the $Z^*$ were below 0 °C until the end of the IPY reference period in 2009.

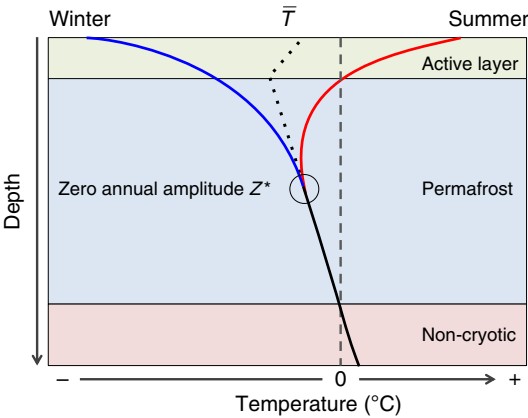

**Fig. 7** Thermal regime of permafrost. Schematic showing the maximum (red line) and minimum ground temperature (blue line) during the year, and their convergence to give the mean annual ground temperature $\bar{T}$ at the depth of zero annual amplitude $Z^*$. Black dots show the schematic mean temperature for permafrost soils. Compiled guided by French[53]

**Compiling permafrost temperature data**. Permafrost temperature data are assembled in the Global Terrestrial Network for Permafrost (GTN-P) Database[16]. They are then transferred to a global data set after a 1-year embargo to allow authors to publish their local findings first. Within the GTN-P Data Management System the data presented were harmonized, quality checked and filtered to generate a standardized global permafrost borehole data set. Data standardization was performed during data entry into the database following international geospatial metadata standards ISO 19115/2 and TC/221. The data management system is based on an object-oriented data model, accessible online at http://gtnpdatabase. org. The GTN-P mean annual ground temperature $\bar{T}$ compilation is accessible online at https://doi.org/10.1594/PANGAEA.884711.

A total of 154 boreholes with 1264 $\bar{T}$ values were used in this study. Data analyses of decadal permafrost temperature change were based on 123 boreholes and 1033 $\bar{T}$ values calculated from >$10^5$ sensor observations.

**Calculating permafrost temperature change**. We used the R environment[44] to calculate the mean permafrost temperature change for every borehole from quality-filtered $\bar{T}$ data. The same measurement depth was used each year for a borehole. The depth was chosen to be the nearest available sensor to the depth of $Z^*$, the depth at which seasonal changes in temperature are ≤0.1 °C (Fig. 7). The nearest depth to $Z^*$ was detected by either an algorithm calculating the difference between annual maximum (summer) and minimum (winter) temperature in the original data starting from the shallowest depth downwards and using cubic spline interpolation between thermistors and a threshold set to sensor accuracy, or by visual inspection of annual maximum and minimum temperature measurements plotted versus depth (Fig. 7). Because the depth of $Z^*$ varies over time as temperature changes, we used an average estimated for the observation period. The data revealed that 19.5% of measurements were from above $Z^*$. 59.8% of measurements represented $Z^*$ and 20.7% were from below $Z^*$. Measurements from boreholes that had no reliable indication of $Z^*$ had a mean depth of 17.1 m, which is well below the average of all indicated $Z^*$ values (mean 14.1 m, median 12 m). Thus, the data distribution represents an approximation to $Z^*$ which minimizes the potential bias caused by seasonal fluctuations.

We created a data set that reflects long-term climate change and avoids large temperature fluctuations caused by seasonal phenomena, e.g., in Antarctica, by excluding data from shallow boreholes that did not reach $Z^*$. Because $Z^*$ could not be determined in all boreholes the minimum depth was set to 10 m. However, five boreholes with depths between 6.7 m and 10 m were included (GTN-P ID's[16]: 137, 860, 861, 877, and 1192), because their depths were equal to $Z^*$, and seasonal fluctuations were less than the instrument precision and accuracy. Boreholes that fulfilled the quality criteria but were not included in this analysis due to depth constraints, represented 22.6% of the original data set. 8.6% were excluded from the Arctic continuous data set; 23.4% from the Arctic discontinuous data set; 30.0% from the mountain data set; and 57.1% from the Antarctic data set. Statistically indifferent temperature trends of the remaining shallow (≤12 m) and deeper (>12 m, max. 40 m) boreholes in the utilized data set confirm that the observed depths near $Z^*$ (Fig. 6b) provide a representative sample tracking climate variability coherently.

We applied different methods to extract information on permafrost temperature changes in single years, in single boreholes and for decadal changes in the permafrost regions, described as follows: We define a set $i = \{2007, \ldots, 2016\}$ to identify the years. To identify the boreholes b we use the GTN-P Database ID. Continuous (full-year) records started at a large number of borehole sites in 2008, the second year of the $4^{th}$ International Polar Year (IPY). To base the reference period for the annual departure calculation on the largest possible number of boreholes we exclude 2007 and estimate the annual differences in $\bar{T}$ in year $y \in i$ and borehole b as

$$\Delta\bar{T}_{y,b} = \bar{T}_{y,b} - 1/2\left(\bar{T}_{2008,b} + \bar{T}_{2009,b}\right) \qquad (1)$$

The last term on the right-hand side of Eq. (1) serves as our mean value for the reference period. We compare this reference period to the latest available mean value period and calculate$\Delta\bar{T}_b$ to rank total temperature differences among boreholes.

$$\Delta\bar{T}_b = 1/2\left(\bar{T}_{2015,b} + \bar{T}_{2016,b}\right) - 1/2\left(\bar{T}_{2008,b} + \bar{T}_{2009,b}\right) \qquad (2)$$

Equations (1) and (2) require data to be available in each of the observation years.

To calculate the rate of temperature change per decade we follow a third approach using the primary mean annual ground temperature data set $\bar{T}_b$ for all available years in $i$ and perform linear regression, according to the following attribution of our data in the regression equation:

$$\bar{T}_b^{\text{reg}} = a_b + c_b x \qquad (3)$$

where $\bar{T}_b^{\text{reg}}$ is the regression estimate of $\bar{T}_b$, $a_b$ is the vertical intercept (the starting temperature in a borehole), $c_b$ is the slope of the regression line, and $x$ is the range of years involved.

The requirement to perform linear regression on b was that $i$ included at least one value $y$ in the IPY period (2007, 2008, or 2009), one value in the modern reference period (2015 or 2016) and a minimum of five values in total. We calculated the rate of temperature change in each borehole as the slope of the linear regression $c_b$ using the linear model function (lm) in the R environment. To generate decadal change values, we extrapolated 37.7% of the borehole data in the Arctic continuous zone, 47.3% in the Arctic discontinuous zone, 29.3% in the mountain zone and 100% in Antarctica for 1–3 years.

The consistency of temperature time series in boreholes depends on sustained data collection at remote sites. At some boreholes, instrumentation was destroyed, damaged or malfunctioned leading to interruptions in data collection[45]. To avoid broken data runs affecting the annual means, measurements at frequencies greater than monthly (e.g. daily or hourly), were aggregated to monthly means before calculating annual means. Mean annual values were based on at least monthly primary data. Data points based on fewer than one measurement every month were allowed only if the sensor depth was equal to or below the depth of zero annual amplitude. Annual means were calculated from original measurements as calendar-year means in the GTN-P Database. Meteorological years in permafrost areas depend on the onset and termination of the freezing and thaw periods, and in previous studies varied spatially. We therefore indicated the starting month of the period in the data set. Mean values contain only the available valid $\bar{T}$ data in each year, and thus the number of borehole temperatures included in change-rate calculations varies between years.

To evaluate temperature changes in the Arctic continuous and discontinuous permafrost zones, in the mountain permafrost and in permafrost in Antarctica, we applied a spatial de-clustering prior to calculating mean values of temperature changes from the boreholes. The spatial de-clustering reduces the bias in the calculation of means caused by an inhomogeneous (clustered) spatial distribution of the boreholes. We grouped the boreholes into ten world zones (Fig. 8) and defined the areas underlain by permafrost by correlating the boreholes with the International Permafrost Association (IPA) permafrost zones[46]. Arctic continuous permafrost represents the mean of four different zones: Arctic continuous permafrost West ($2.41 \times 10^6$ km$^2$), Arctic continuous permafrost West islands ($1.57 \times 10^6$ km$^2$), Arctic continuous permafrost Europe ($0.22 \times 10^6$ km$^2$), and Arctic continuous permafrost East (Asia) ($6.62 \times 10^6$ km$^2$). Arctic discontinuous permafrost is averaged over three zones: Arctic discontinuous permafrost West ($3.91 \times 10^6$ km$^2$), Arctic discontinuous permafrost East (Asia) ($3.86 \times 10^6$ km$^2$), and Arctic discontinuous permafrost Europe ($0.28 \times 10^6$ km$^2$). Mountain permafrost is averaged over two zones: Chinese mountains ($2.07 \times 10^6$ km$^2$), and Other mountains ($2.33 \times 10^6$ km$^2$) including the Alps and other sites with high elevations >1000 m a.s.l. such as in Scandinavia and the North American Cordillera. Antarctica is treated as one zone ($0.05 \times 10^6$ km$^2$ [6,47]). For comparing temperature trends between North American and north Asian permafrost we define two separate data sets by excluding southern, European, and central Asian boreholes. Within the zones, clusters of boreholes close together were grouped when the sum of longitude and latitude differences were <0.1 decimal degree and the $\bar{T}$ values of adjacent boreholes were averaged before calculating the mean temperature change.

To estimate the mean annual temperature change in each zone we applied area-weighted arithmetic averaging of $\bar{T}$ values in boreholes. To preserve the signal of local outlier trends showing atypical temperature change directions and magnitudes (e.g., in parts of Antarctica and in Québec, Canada), we did not use medians. To suppress near-surface and geothermal changes indices of boreholes were distributed as three possible integers to multiply the sites before averaging, according to the following criteria: (i) $\bar{T}$ is available in each year of the reference periods indicated in Eqs. (1) and (2), and (ii) $\bar{T}$ depth is equal to the depth of $Z^*$ and >10 m (few exceptions were made according to the depth of $Z^*$ as described above).

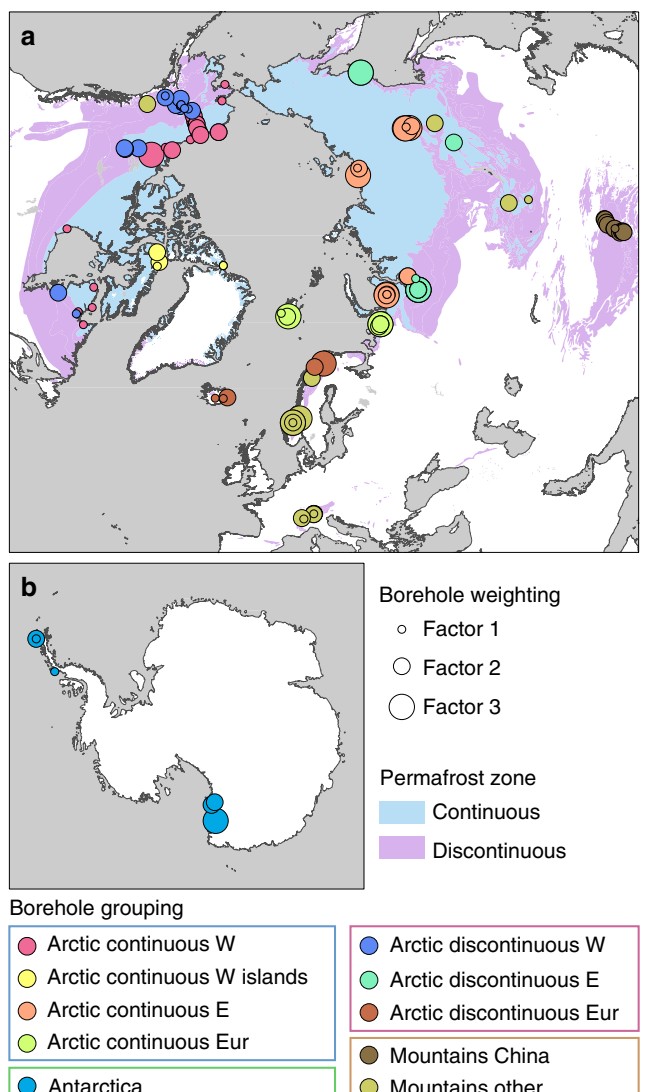

**Fig. 8** Weighting and grouping of boreholes. Map showing the indices and zoning of boreholes prior to area-weighting and calculation of mean temperature changes. **a** Northern Hemisphere. **b** Antarctica. Permafrost zones are derived from the International Permafrost Association (IPA) map[46]. World Borders data are derived from http://thematicmapping.org/downloads/world_borders.php and licensed under CC BY-SA 3.0 (https://creativecommons.org/licenses/by-sa/3.0/)

**Calculating air temperature change**. The set of air temperature data monitored at borehole sites is incomplete. To develop data comparable to the permafrost temperature data, we calculated mean annual air temperatures ($\hat{T}$) from ERA Interim 2 m air temperature data set with 80 km spatial resolution. We derived the reanalysis time series for each borehole from linear interpolation of the four nearest grid points surrounding the borehole coordinates. Mean annual values were calculated from December until November. Given, that the propagation of atmospheric temperature change downward to the depth of $Z^*$ takes up to several years[25,37], depending on the local thermal diffusivity[24], we extended the time series shown in Fig. 4 backwards to 2000 and used the standard reference period 1981–2010 to estimate anomalies.

We define a set $j = \{1981,...,2016\}$ to identify the years being considered. We use the coordinates of boreholes $b$ defined in Eq. (1) and calculate the annual difference for specific years $y \in j$ in $\hat{T}$ as

$$\Delta \hat{T}_{y,b} = \hat{T}_{y,b} - \frac{1}{30} \sum_{j=1981}^{2010} \hat{T}_{j,b} \qquad (4)$$

Based on the average propagation of surface temperature towards $Z^*$ of 4 years[25] we calculated 4-year end-point running means to compare air temperature with permafrost temperature changes. To calculate the rate of temperature change over a

decade, we apply linear regression on $\hat{T}_{y,b}$ for all $y \in j$ using the linear model function in the R environment and the slope of the linear regression in an annual array between 2004 and 2016 and multiplied the annual change rates by 10. Data analyses of air temperature change were based on 137 borehole sites and 4932 $\hat{T}$ values.

**Calculating snow thickness change**. We calculated the mean annual snow thickness ($\hat{S}$) for the Arctic continuous and the discontinuous permafrost zone from the Canadian Meteorological Centre (CMC) daily snow depth analysis data with 24 km spatial resolution[48]. We derived the reanalysis time series for each borehole from linear interpolation of the four nearest grid points surrounding the borehole coordinates. Mean values were calculated from December until February for each year in the data set. To identify winters we use subsequent years, e.g. in the time series we assign the 1999–2000 winter to 2000.

Given that 1999 is the earliest available year in the data set we define a set $k = \{1999,...,2016\}$ to identify the winter years, where $y \in k$. We use the coordinates of boreholes $b$ defined in Eq. (1) and calculate the annual difference in $\hat{S}$ as

$$\Delta \hat{S}_{y,b} = \hat{S}_{y,b} - \frac{1}{12} \sum_{k=1999}^{2010} \hat{S}_{k,b} \qquad (5)$$

The onset snow has an impact on the ground thermal regime. To assess the onset of snow cover, we assemble a set of snow depths dates at daily resolution between 1 September and 30 April in a set of days $l = \{1,2,3,...,242\}$ for every year in $k$. In leap years $l = \{1,2,3,...,243\}$. To calculate the onset date of snow SO we use the first day $d_{k,b}^{SO}$ reaching 6 cm in $l$ for which the following 5 days, adding up to a synoptic time scale of 6 days, retain a daily snow cover of at least 6 cm[49].

The insulation maximum of snow SIM is reached when the snow cover accumulated to a thickness between 40 and 50 cm[33,37]. Accordingly, we set SIM based on the first day $d_{k,b}^{SIM}$ in $l$ reaching 50 cm, or, if it is not reached, take the day representing the maximum snow cover in $l$ (below 50 cm).

To assess the end of snow cover SE, we assemble the snow depth dates at daily resolution between 1 September and 30 August in a set $m = \{1,2,3,...,365\}$ for every year in $k$. In leap years $m = \{1,2,3,...,366\}$. To calculate SE we use the first day $d_{k,b}^{SE}$ in $m$ reaching down to less than 1 cm after a decreasing gradient of at least 8 cm over 6 days, or, if this gradient is not reached, the first day of at least 6 subsequent snow free (<1 cm) days.

**Measurement accuracy**. The reported measurement accuracy of our temperature observations, including manual and automated logging systems, varied from ±0.01 to ±0.25 °C with a mean of ±0.08 °C. Previous tests have shown the comparability of different measurement techniques to have an overall accuracy of ±0.1 °C[3]. Thermistors are the most commonly used sensors for borehole measurements. Their accuracy depends on (1) the materials and process used to construct the thermistor, (2) the circuitry used to measure the thermistor resistance, (3) the calibration and equation used to convert measured resistance to temperature, and (4) the aging and resulting drift of the sensor over time. Thermistors are typically calibrated to correct for variations due to (1) and (2). About 20% of the boreholes are visited once per year and measured at or below $Z^*$ using single thermistors and a data logger. In this case the system is routinely validated in an ice-bath allowing correction for any calibration drift. The accuracy of an ice-bath is ~± 0.01 °C[50]. Using the offset determined during this validation to correct the data greatly increases the measurement accuracy near 0 °C, an important reference point for permafrost. The remaining systems are permanently installed and typically ice-bath calibrated at 0 °C before deployment. The calibration drift is difficult to quantify as thermistor chains are not frequently removed for re-calibration or validation. In many cases removal of thermistor chains becomes impossible some time after deployment, e.g. because of borehole shearing.

The drift rate among bead thermistors from different manufacturers was <0.01 °C per year during a 2 year experiment at 0, 30, and 60 °C[51]. The calibration drift of glass bead thermistors was found to be 0.01 mK per year[52], at an ambient temperature of 20 °C. A single drifting thermistor in a chain is detectable through its anomalous temporal trend. Such data were excluded from our data set. The absolute accuracy of borehole temperature measurements, in terms of their representativeness of the temperature distribution in undisturbed soil, also depends on the depth accuracy of the sensors' positions in the borehole. This study is concerned with temperatures at $Z^*$, where temperature gradients are typically small (<0.1 °C m$^{-1}$). Consequently, mm-level positioning accuracy does not significantly impact measurement accuracy. Finally, as this study is concerned with annual averages, adequate chronometry is ensured.

The above discussion of accuracy relates to the absolute temperature values measured, but the detection of temperature change is more accurate because errors in calibration offset have no impact, sensor nonlinearities are generally small and not of concern. We therefore consider <0.1 °C a conservative average estimate of the accuracy of temperature change on an individual sensor basis.

**Confidence intervals and statistical significance**. Permafrost and air temperature departure from 2008 until 2016 ($\Delta \bar{T}_{i,b}$ and $\Delta \hat{T}_{y,b}$) and the regression from 2007 until 2016 of each borehole were used to calculate the 95% confidence intervals within each world zone using a Student $t$-test in the R environment (52% $p < 0.05$, 48% $p > 0.05$, mean $|t| = 3.4$). The upper and lower confidence boundaries

were calculated from de-clustered and indexed boreholes. Mean confidence intervals for composite permafrost zones (global, Arctic continuous, Arctic discontinuous, mountain, Asian and American) were area-weighted. Antarctica consists of one zone and thus area-weighting is not applicable. Given a non-normal, unimodal, and only slightly skewed distribution of data in similarly shaped subsets (regions) gained by eq. 3 (Figs. 5, 6), we performed a Wilcoxon Signed-Rank test and a Kruskal–Wallis test to assess the significance of the difference to zero and the differences between medians, respectively. To consider false positives, we performed a False Discovery Rate adjustment of the p-values, resulting in 43.3% $p < 0.05$, 56.6% $p > 0.05$, median 0.08 in a data matrix of 9 years (eq. 1) versus 10 permafrost world zones indicated in Fig. 8. Boxplots represent 25–75% quartiles and whiskers are 1.5 interquartile ranges from the median.

## Data availability

The GTN-P global mean annual ground temperature data for permafrost near the depth of zero annual amplitude (2007–2016) is accessible online at https://doi.org/10.1594/PANGAEA.884711.

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

## Acknowledgements

This research would not have been possible without the long-term commitment of all observers to site maintenance, data collection, and their willingness to share permafrost borehole data. All data were compiled by the Global Terrestrial Network for Permafrost (GTN-P). We thank the International Permafrost Association for financial support. We thank Jerry Brown for initiating the borehole metadata collection and Christina Roolfs for mathematical review. This research was supported by grants from (in alphabetical order) AGAUR ANTALP #2017-SGR-1102 (Catalonia); BMBF PALMOD #01LP1510D (Germany); ERC PETA-CARB #338335 (EU); FCT #PERMANTAR2017-18/PROPOLAR (Portugal); Formas #214-2014-562 (Sweden); HGF COPER #VH-NG-801 (Germany); Horizon 2020 Nunataryuk #773421 (EU); JSPS KAKENHI #25350416, #21310001 (Japan); MESC #RFMEFI58718X0048, #14.587.21.0048-SODEEP (Russia); MeteoSwiss in the framework of GCOS Switzerland, FOEN and SCNAT for the Swiss Permafrost Monitoring Network PERMOS (Switzerland); Natural Resources Canada; NNSF #41690144, #41671060 (China); NRC TSP #176033/S30, #157837/V30, #176033/S30, #185987/V30 (Norway); NSERC #2014-04084, #2015-05411 (Canada); NSF OPP #1304271, #1304555 #1836377; ICER #1558389, #1717770 (USA); PNRA #16_00194 (Italy); Ramon y Cajal #RYC-2015-17597 (Spain); RAS PP #15, #51, #55, GP #AAAA-A18-118022190065-1, #18-218012490093-1 (Russia); RFBR #18-05-60004, #18-55-11003, #16-05-00249, #16-45-890257-YaNAO, #18-55-11005 AF_t(ClimEco), #18-05-60222-Arctica (Russia); RSCF #16-17-00102 (Russia); National Research Foundation, SNA #14070874451 (South Africa).

## Author contributions

The study was initially conceived during a GTN-P workshop in 2015. B.K.B. led the analyses and writing of the manuscript. S.L.S., J.N., H.M., G.V., D.S., P.S., V.E.R. and A.G.L. are principal co-authors. A.A., M.A., J.B., W.L.C., H.H.C., B.D., R.D., D.D., B.E., G.G., M.G., T.I.-N., K.I., M.I., M.J., D.K., A.K., P.K., H.L., C.L., D.L., G.M., I.M., N.M., M.O., M.P., M.R., A.B.K.S., D.S., C.S., P.S., A.V., Q.W., K.Y. and M.Z. contributed with data collection and expert assessment of borehole data. H.J., A.J., T.K. and J.-P.L. performed database coding, data processing, and data analyses. All authors contributed to analysis of the results and revision of the manuscript.

## Additional information

**Competing interests:** The authors declare no competing interests.

Boris K. Biskaborn[1], Sharon L. Smith[2], Jeannette Noetzli[3], Heidrun Matthes[1], Gonçalo Vieira[4], Dmitry A. Streletskiy[5], Philippe Schoeneich[6], Vladimir E. Romanovsky[7], Antoni G. Lewkowicz[8], Andrey Abramov[9], Michel Allard[10], Julia Boike[1,11], William L. Cable[1], Hanne H. Christiansen[12], Reynald Delaloye[13], Bernhard Diekmann[1,14], Dmitry Drozdov[15], Bernd Etzelmüller[16], Guido Grosse[1,14], Mauro Guglielmin[17], Thomas Ingeman-Nielsen[18], Ketil Isaksen[19], Mamoru Ishikawa[20], Margareta Johansson[21], Halldor Johannsson[22], Anseok Joo[22], Dmitry Kaverin[23], Alexander Kholodov[7,9], Pavel Konstantinov[24], Tim Kröger[25], Christophe Lambiel[26], Jean-Pierre Lanckman[22], Dongliang Luo[27], Galina Malkova[15], Ian Meiklejohn[28], Natalia Moskalenko[15], Marc Oliva[29], Marcia Phillips[3], Miguel Ramos[30], A. Britta K. Sannel[31], Dmitrii Sergeev[32], Cathy Seybold[33], Pavel Skryabin[24], Alexander Vasiliev[15,34], Qingbai Wu[27], Kenji Yoshikawa[7], Mikhail Zheleznyak[24] & Hugues Lantuit[1,14]

[1]Alfred Wegener Institute Helmholtz Centre for Polar and Marine Research, Potsdam 14473, Germany. [2]Geological Survey of Canada, Natural Resources Canada, Ottawa ON-K1A 0E8, Canada. [3]WSL Institute for Snow and Avalanche Research SLF, Davos CH-7260, Switzerland. [4]CEG/IGOT, Universidade de Lisboa, Lisbon 1600-276, Portugal. [5]George Washington University, Washington DC 20052, USA. [6]Institut de Géographie Alpine, Grenoble F-38100, France. [7]University of Alaska Fairbanks, Fairbanks AK-99775, USA. [8]University of Ottawa, Ottawa K1N 6N5, Canada. [9]Institute of Physicochemical and Biological Problems of Soil Science, RAS, Moscow 142290, Russia. [10]Université Laval, Centre d'études nordiques, Québec G1V 0A6, Canada. [11]Humboldt-Universität, Geography Department, Berlin 10099, Germany. [12]The University Center in Svalbard, Longyearbyen N-9171, Norway. [13]University of Fribourg, Fribourg CH-1700, Switzerland. [14]University of Potsdam, Potsdam 14469, Germany. [15]Earth Cryosphere Institute, Tyumen Scientific Centre SB RAS, Tyumen 625000, Russia. [16]University of Oslo, Department of Geosciences, Oslo N-0316, Norway. [17]Insubria University, Department of Theoretical and Applied Sciences, Varese 21100, Italy. [18]Technical University of Denmark, Department of Civil Engineering, Kgs. Lyngby DK-2800, Denmark. [19]Norwegian Meteorological Institute, Oslo 0313, Norway. [20]Hokkaido University, Sapporo 060-0810, Japan. [21]Lund University, Lund 22362, Sweden. [22]Arctic Portal, Akureyri 600, Iceland. [23]Komi Science Centre, RAS, Syktyvkar 167972,

Russia. [24]Melnikov Permafrost Institute, RAS, Yakutsk 677010, Russia. [25]Free University Berlin, Geography Department, Berlin 12249, Germany. [26]University of Lausanne, Lausanne 1015, Switzerland. [27]Northwest Institute of Eco-environment and Resource, CAS, Lanzhou 730000, China. [28]Rhodes University, Grahamstown 6140, South Africa. [29]University of Barcelona, Barcelona 08001, Spain. [30]Universidad de Alcalá, Madrid 28801, Spain. [31]Stockholm University, Stockholm SE-106 91, Sweden. [32]Institute of Environmental Geoscience, RAS, Moscow 101000, Russia. [33]National Soil Survey Center, Lincoln NE-68508, USA. [34]Tyumen State University, Tyumen 625003, Russia

