## [Peer Review File · Nature Communications]

Reviewers' comments:

Reviewer #1 (Remarks to the Author):

Review of Global permafrost temperatures on the rise by B.K. Biskaborn et al. Review by C.R. Burn.

A global summary of recent trends in permafrost temperatures will be of widespread interest in the scientific, engineering, and environmental assessment communities, especially with respect to practitioners concerned with the polar regions and those interested in the effects of climate change. The most well-resourced research on permafrost at present concerns the ultimate destination of the carbon stocks presently entombed in frozen ground. There is sincere and genuine concern that decomposition of permafrost carbon will enhance the greenhouse effect significantly and place further unwanted pressure on the global atmospheric carbon budget. Changes in permafrost conditions are critical to estimating the rate and magnitude of these effects.

The GTN-P is a project that has been in progress for several years, with the explicit goal of producing summary statements such as the paper under review. The data base has considerable value, like the snapshots of permafrost state at global and continental scale presented after the International Polar Year. But this paper is, indeed, the first to report on change over time throughout these broad regions. For these reasons I believe it has considerable scientific merit. However, my reading of the paper has brought out several points that may require further consideration. In particular, I do not consider that the evidence presented makes a clear connection between change in air and permafrost temperatures, especially for the period 2007-14. The number and nature of these concerns do not allow me to recommend that the paper be published in its present form.

1. P. 1, l. 19. "Permafrost warming ... can unlock the organic carbon in frozen sediments." Really? I doubt if a change in permafrost temperature from -8 to -6 °C or even -4 to -2 °C will release carbon from permafrost in materially significant quantities. The microbes responsible for carbon release are not particularly effective at these temperatures. Thawing of permafrost and deepening of the active layer may have just such an effect, but no data is presented in this paper on active layer depths. And in that case, the release is not from permafrost but from seasonally unfrozen ground.
2. p. 1, l. 21 A minor point. The papers published by many of the authors in 2010 were intended to present a summary and assessment of permafrost at circumpolar scale. The values summarized in those compilations may not have all come from the same depth, but it is misleading to suggest that either those summaries were fundamentally inconsistent and flawed, or, alternatively, that they do not represent global-scale assessments.
3. P. 1, l. 27. Another minor point. A decade has not yet passed since 2007-09. The values presented in units of per decade are therefore extrapolations.
4. P. 1, l. 32. Fig. 3 presents plots of ground temperature increases and changes in air temperature for various regions in 2008-2016. The ground temperatures indicate ground warming over this period. The air temperature records only demonstrate warming in the last two years (2014-16). From 2007-14 there is no consistency in the data. They go up and down, with overall change of about 0 °C with respect to conditions in 2007-09. How does "permafrost warming reflect the Arctic and high elevation amplification of 2-m air temperature increase" when there was no increase from 2007-14? Why did the ground warm between 2007-14 if the atmosphere was not doing so? I am sure there is a simple explanation but I cannot find it in the paper, and I expect it requires more data than are presented in this paper. There are suggestions in lines 116-120, but no definitive reason is provided. I expect the authors are looking in the wrong direction, and should think about conditions below the point of observation. If the warming is amplified in the regions of interest, it is particularly awkward that no increase in temperature occurred, except for 2014-16.
5. P. 2, l 51. "This process will further accelerate". The values presented in this sentence are projections based upon modelling that invokes a large number of assumptions about (a) the lability of carbon in permafrost, (b) the depth distribution of the carbon, (c) the rate of active layer deepening. The data presented are for 80 years and 280 years away. I do not doubt the integrity of the scientists, but they have provided projections, and those will change as climate change progresses. To state the effects of carbon amplification with the language of quantitative certainty does not permit revision. The effects may be much greater.

6. P. 2, l 66. If all the data were available for 2007-16 I would accept this statement. But p. 5, l 220 states that data for 2007 were excluded! In fact, the initialization was for the average of data from 2008 and 2009 (l. 78). The end of the series was the average of 2015 and 2016. If the period begins in early 2009 and ends in early 2016, only 7 years of time are involved, not a decade as stated in line 66.
7. P. 2, l.70. The depth of zero annual amplitude (Z^*) may be determined when the timescale of temperature fluctuation is one year. If surface temperature is changing over a longer timescale, as with these data, then Z^* is difficult to determine and will vary with the rate of long-term temperature change. This is a considerable problem, because the specific depth at which seasonal effects “are negligible” will vary with ground materials and with atmospheric trends. In practice, these considerations mean that Z^* will be overestimated. This creates another consideration, because the rate of temperature change declines with depth. It is then very difficult to compare sites, if they are being examined at a range of depths. I do not know how to circumvent this problem, except by taking an arbitrary depth (e.g. 15 m) with sufficiently frequent observations that a reliable annual mean may be calculated. The depth may need to be different in the various permafrost environments/zones discussed.
8. P. 2 l. 75. The authors state that permafrost thaw occurred at five boreholes. The significance of the statement is not clear. Did the active layer only deepen in five boreholes out of 124? If the active layer deepened in the majority of boreholes, then permafrost thaw occurred in the majority. Does this statement mean, instead, that permafrost was eradicated in five boreholes?
9. P. 2, l. 93. What constitutes the eastern Arctic and what forms the western Arctic? These are relative terms and for a circumpolar phenomenon, they vary with location of the observer. Eurocentric views might have been the norm two decades ago, but now, the frames of references are much expanded, especially considering the number of scientists – readers of the Nature journals - in China.
10. P. 3, l. 114. The lack of correlation on an annual time scale is not surprising, given comment 4 and the data displayed in Fig. 3. There is some merit to point 1 (line 116), but the principal problem is that from 2007-14 the reanalysis data do not show any trend. Point 2 (line 118) implies, in this context, that there has been a widespread increase in snow accumulation. There must be some data available from the North American Arctic to demonstrate this, but I have not seen it in my field area (Canada’s western Arctic coast and Yukon). Point 3 is also interesting, but fundamentally, we would expect various rates of change in ground temperature in response to a change in air temperature from these factors. The difficulty is that Fig. 3a shows increases in ground temperature from 2008-14, when no air temperature increase took place.
11. P. 3, l. 135. Warming of permafrost in high mountains and in the High Arctic would reflect atmospheric warming well if there had been warming in 2007-14. Instead some other explanation is required. There are many places in the discontinuous permafrost zone that respond rapidly to climate warming, but they are mostly in bedrock. Relatively few boreholes have been drilled in bedrock outcrops in continental discontinuous permafrost that are analysed by scientists, because these require industrial budgets.
12. P. 4, l. 149, p. 159. The term “deep permafrost” needs considerable further thought. Lines 159 ff suggest that for these authors “deep permafrost” is a few m thick, because this is where most permafrost carbon is found. This is inappropriate. In Russia and northern North America, permafrost may be well over 400 m thick. We consider deep permafrost to be ground several 100s of m below the surface. Permafrost in the upper few m of the ground is by no means deep. Even the 30 m or more of yedoma sediments found in Alaska and Russia would not be considered “deep” by most scientists and engineers who work in the Arctic.
13. P. 5, l. 211. Shallow boreholes, according to the text are less than 10 m deep, but deep permafrost may not be. Or is the carbon referred to in line 149 more than 10 m below the surface?
14. P. 4, l. 214. It would be useful to know how much of the permafrost zone has been excluded by the decision to set the minimum depth of temperature record to 10 m. Does this criterion, which means that few data are considered from permafrost thinner than 10 m, lead to 25% of the permafrost zone being unrepresented in the statistics, or is the proportion of the area different?

- Fig. 1. There are 17 locations in part a and 14 lines recognizable in b. The colours indicate that the ground temperatures in the Swiss Alps are comparable to the northern coast of Greenland/Ellesmere Island.
- Fig. 2. The coldest permafrost in map a is in the mountains of Alberta. This is a surprising result, not well known before. Is permafrost there thicker than on Ellesmere Island?
- Fig. 3 has been considered at length in this review.
- Fig. 5. The mean temperature within the active layer characteristically decreases with depth due to seasonal change in thermal properties (the thermal offset). Why has this not been shown on the diagram? Does this diagram imply that most boreholes under consideration are in bedrock or dry ground? The thermal offset is prevalent in ground with high water content, where the “latent heat effect” (line 138) is important. The unfrozen ground below permafrost is not commonly considered to be talik, unless, of course, there is frozen ground below this layer.
- Fig. 6. It is confusing to indicate discontinuous permafrost in the same colour as the boreholes in the continuous zone for North America.

Reviewer #2 (Remarks to the Author):

This manuscript describes a collection of boreholes for permafrost temperature monitoring. The authors describe the network and provide initial results on recent temperatures and temperature trends. The analysis is straightforward, for the most part, and includes a nice discussion on spatial de-clustering and measurement error. Overall, I found this paper suitable for publication. This dataset will provide a valuable constraint for models. I have a few minor suggestions for improvement.

1. I would change “will” to “could” on line 52 since the modeling of the permafrost climate-carbon feedback remains poorly constrained and models do not show strong agreement.
2. I would say that permafrost is not yet adequately integrated into MOST ESMs. Several groups have been working to incorporate permafrost processes into their models. I would also say that the lack of permafrost temperature observations is just one of the reasons why permafrost processes have not been incorporated. Another reason, for example, is that the deep soils needed for permafrost modeling require long (and costly) spinup times. In general, from my perspective, the advantage of having observed permafrost temperatures and their trends is that they provide a way to evaluate the skill of models in terms of their ability to reproduce the permafrost temperatures and their trends.
3. Reading this through raised a question for me. Are there borehole measurements outside of the permafrost domain? If so, it might be worth at least a brief discussion of these datasets here. To provide a good test of an ESM, it would be preferable to examine trends within and outside of the permafrost domain, which would allow researchers to assess the uniqueness (or not) of the permafrost zone with respect to soil temperature trends.
4. The last sentence (or two) in the methods section seems important. “Enhancing existing monitoring sites through co-location with meteorological stations would also facilitate improved understanding of microclimate and buffer layer influences and also provide the data necessary for a full assessment of permafrost responses to ongoing climate change.” Perhaps this sentence or something akin to it should be incorporated into the main text.

Reviewer #3 (Remarks to the Author):

This paper reports a synthesis of an important, global data set of mean permafrost temperatures that will of interest to a wide audience of researchers, managers, and policy-makers. The synthesis is original and likely to influence thinking in the field of climate science, especially in polar regions. However, there are a number of unclear and inaccurate statements that need to be addressed before this paper is acceptable for publication in Nature Communications.

In my comments below "L#" refers to the line number(s) in the paper. I have attached an annotated copy of the PDF which the authors might find useful.

The most important points requiring clarification are:

L30-31: The global mean permafrost value reported in the abstract appears to be an arithmetic mean. I think a mean weighted by the areas of the defined regions would be more appropriate. Are these means different?

L154-155: According to Fig. 2c, it seems that there are some borehole sites at which cooling has occurred. Although uncommon, this does not support the statement that the "entire zone" of continuous permafrost in the Arctic is warming. Furthermore, across all of the boreholes, a substantial portion ($(124-71)/124 * 100 = 43\%$) either cooled (12) or did not warm or cool ($124-71-12 = 41$). This is mentioned briefly at lines 74-75. I think a more balanced statement is warranted in the abstract. For example, "While cooling was observed in some boreholes and others did not demonstrate significant warming over the decade, the overall mean temperatures in all 10 regions increased significantly."

L220: I understand that it is common practice in papers on climate change to refer to temperature "anomalies". These are usually understood to mean departures from a long-term and well-established norm and are interpreted to be unusual. I fully understand that the authors think the departures that they report here are unusual, and I agree. However, in this case, the departure is from an arbitrary, two-year mean. I think it is more accurate to refer to these as "departures" rather than "anomalies".

L246-247: How many monthly measurements were required from a mean T value to be valid? Was it necessary for there to be at least one value in each of 12 months? If not, why not? See also the comment at lines 250-253.

L247-249: The sensor selection criteria noted in lines 203-204 is important here. At how many sites were monthly data missing for boreholes with sensors below Z^* ? Am I correct that <no> borehole data were included for sensors that were shallower than Z^* and had missing monthly data?

Lines 250-253: It appears from these statements that the data reported here are based on temperature measurements made over different numbers of months depending on the local meteorological conditions (i.e., latitude and elevation). Specifically, it appears that the permafrost means are not annual means, but a mean based on a variable portion of the year, depending on the site. Thus, the mean permafrost temperature at sites could be influenced by different seasonal characteristics. If the primary focus of this paper is on temperature change or departures from a reference, why not define that reference to be as consistent as possible? For example, select the three months that are most indicative of boreal or austral "winter" conditions?

Lines 250-253: It appears from these statements that the data reported here are based on temperature measurements made over different numbers of months depending on the local meteorological conditions (i.e., latitude and elevation). Specifically, it appears that the permafrost means are not annual means, but a mean based on a variable portion of the year, depending on the site. Thus, the mean permafrost temperature at sites could be influenced by different seasonal characteristics. If the primary focus of this paper is on temperature change or departures from a reference, why not define that reference to be as consistent as possible? For example, select the three months that are most indicative of boreal or austral "winter" conditions?

L284-291: It is not clear whether the numbers preceding each weighting criteria are in fact the weights or are simply index numbers. This should be clarified. How were the weights actually used in this assessment?

L329-332: It is unclear to me what values have been assessed in the matrix of t-tests provided as supplementary data. I presume these are the permafrost anomalies, but this should be stated explicitly. If the data are not normally distributed, why use a t-test? A non-parametric test (i.e. Wilcoxon Signed-Rank test) might be more appropriate. Furthermore, there are 100 individual tests in this table. Simple probably suggests that perhaps 20% of these would be "false positives." Thus, in assessing this matrix of results, some sort of Bonferroni-type adjustment should be included. If this supplemental data table is to be included with the final manuscript, the p values should be reduced to no more than 3 significant digits. There is little meaning to a p value with 7 significant digits.

L322-323: Why arbitrarily pick a mean accuracy that you know is lower than the accuracy of some sensors in the network? It seems that the accuracy should be set nearer the least accurate sensor in the network. If the accuracy of +0.25°C is anomalously high, that should be stated. Or some other criteria other than "arbitrary" should be given to justify this important threshold.

The following are a number of edits I suggest to the authors to help clarify other aspects of the paper:

L58: Need a better transition here: "which" rather than "these"?

L71-73: The rates are not "decadal"; rather, they are "rates calculated over a decade".

L105: Were these really "analogue" calculations? The means appear to have been calculated using digital data.

L122: "de-noising" is an awkward term. Rather, "acts as a filter that removes rapid variation caused by..."? Is this variation really noise (i.e., error) or real variation that obscures the long-term trends the authors want to reveal?

L126: "rates of change"

L138: The reference here is unclear. Did air temperatures increase 5x faster in the discontinuous zone than air temperatures in the continuous zone? Or did the discontinuous permafrost temperatures increase 5x faster than the permafrost temperatures? Specifically, it is unclear if the final reference to "permafrost temperatures" refers to the permafrost itself or to the air temperatures in that region.

L186-187: Is there a place, either on the GTN-P website or in the supplemental data, where key aspects of the types of harmonization, filtration, and standardization employed to make the data set coherent could be recorded? This might be important for future users of this global data set.

L203-204: How many sites had sensors above Z^* vs below Z^* ?

L208-209: The phrase "estimation of the scientist responsible for the borehole" is vague. What criteria were used to ensure consistency at a site, over time and across the network?

L215: The phrase "showed little seasonal fluctuation" is vague. I assume the seasonal fluctuations were less than the instrument precision and accuracy. If so, this should be stated.

L220: The reason that the 2007 borehole data were excluded should at least be mentioned briefly here. From the methods, I think this was because there were too many missing values in 2007.

L300-301: This is confusing. It appears that only two years (0 and 1) are used to calculate Mean Annual Air Temperatures? However, equation 3 suggests that the entire available record was used. This should be clarified.

L310: To be clear, I suggest adding "the <air> temperature".

L337-347: This assessment of the data seems better placed after line 163, just before the methods section.

L507 (Fig. 1): It is difficult to make out the color gradations for each borehole on the figure and then transfer this to the map. It would be helpful to have the borehole numbers on the figure.

L533-534 (Fig. 3): The temperature confidence intervals as dashed lines are difficult to decipher. The color difference between the Arctic Discontinuous and Mountain data is especially difficult to make out.

L540 (Fig. 4): What are the error bars? SD? SE? Range? Inter-quartiles?

Breck Bowden
University of Vermont

Revision Notes

16 June 2018, Biskaborn et al.

Reviewer #1 and answers:

Review of Global permafrost temperatures on the rise by B.K. Biskaborn et al. Review by C.R. Burn.

A global summary of recent trends in permafrost temperatures will be of widespread interest in the scientific, engineering, and environmental assessment communities, especially with respect to practitioners concerned with the polar regions and those interested in the effects of climate change. The most well-resourced research on permafrost at present concerns the ultimate destination of the carbon stocks presently entombed in frozen ground. There is sincere and genuine concern that decomposition of permafrost carbon will enhance the greenhouse effect significantly and place further unwanted pressure on the global atmospheric carbon budget. Changes in permafrost conditions are critical to estimating the rate and magnitude of these effects.

The GTN-P is a project that has been in progress for several years, with the explicit goal of producing summary statements such the paper under review. The data base has considerable value, like the snapshots of permafrost state at global and continental scale presented after the International Polar Year. But this paper is, indeed, the first to report on change over time throughout these broad regions. For these reasons I believe it has considerable scientific merit. However, my reading of the paper has brought out several points that may require further consideration. In particular, I do not consider that the evidence presented makes a clear connection between change in air and permafrost temperatures, especially for the period 2007-14. The number and nature of these concerns do not allow me to recommend that the paper be published in its present form.

We are grateful for the time and expert knowledge contributed by Dr. Chris Burn in his detailed review. We believe that the concerns raised have been addressed in the revised version of the manuscript. Your review helped a lot to improve the integrity of our permafrost-air comparison and in a number of statements we missed to elucidate in sufficient detail. Here, we respond to your comments and elaborate the modifications we did in the original text and the figures.

1. P. 1, l. 19. *“Permafrost warming ... can unlock the organic carbon in frozen sediments.” Really? I doubt if a change in permafrost temperature from -8 to -6 °C or even -4 to -2 °C will release carbon from permafrost in materially significant quantities. The microbes responsible for carbon release are not particularly effective at these temperatures. Thawing of permafrost and deepening of the active layer may have just such an effect, but no data is presented in this paper on active layer depths. And in that case, the release is not from permafrost but from seasonally unfrozen ground.*

Originally, we intended to say that permafrost warming that finally leads to thawing will have the effect of releasing carbon from previously frozen sediments. The shortness of the abstract did not allow us to be too specific here. However, we agree with your statement and reword to “Permafrost warming has the potential to amplify global climate change, because it can unlock soil carbon when frozen sediments start to thaw”.

2. p. 1, l. 21 *A minor point. The papers published by many of the authors in 2010 were intended to present a summary and assessment of permafrost at circumpolar scale. The values summarized in those compilations may not have all come from the same depth, but it is misleading to suggest that either those summaries were fundamentally inconsistent and flawed, or, alternatively, that they do not represent global-scale assessments.*

Yes, we agree. This sentence was misleading. The so-called snapshot papers provided substantial insights in the permafrost thermal state. What they missed, because it was not possible at that time, is a globally standardized and consistent assessment of the development of permafrost temperature. Moreover, mountain permafrost was poorly represented and Antarctica was not involved in the synthesis. Due to the lack of continuous measurements and standardized data their synthesis paper was based on a time-slice summarizing the three years of the IPY 2007-2009, but during this time no global assessment of the permafrost development. We therefore believe that it is appropriate to reword to “Yet to date, no globally consistent assessment of permafrost temperature change has been compiled.”. We also emphasize the important work of the IPY project lateron: “A key outcome of the IPY project was that the number of accessible boreholes used for temperature monitoring has significantly increased”.

3. P. 1, l. 27. *Another minor point. A decade has not yet passed since 2007-09. The values presented in units of per decade are therefore extrapolations.*

We have clarified this point. The analyses presented are based on a dataset with 10 columns starting from 2007 and ending in 2016, which is one decade. The assessment of single boreholes, based on “anomaly” (departure) calculation

explained in equations 1 and 2 has slightly higher quality criteria and is therefore based on 9 years from 2008 to 2016. This is necessary because many boreholes started temperature logging after 2007, which would then have to be excluded in the ranking. This is also shown in figure 3. The absolute numbers for the main zones presented in the text are properly calculated change rates over an entire decade starting in 2007. However, our quality criteria, also permitted borehole records that cover e.g. 2008 and 2015, and therefore some records represent extrapolations in the regression analysis. We stated in the revised version the scale of extrapolation: “We extrapolated 37.7 % of the boreholes in the Arctic continuous zone, 47.3 % in the Arctic discontinuous zone, 29.3 % in the mountain zone and 100 % in Antarctica accordingly for 1-3 years to generate decadal change values.”.

4. P. 1, l. 32. Fig. 3 presents plots of ground temperature increases and changes in air temperature for various regions in 2008-2016. The ground temperatures indicate ground warming over this period. The air temperature records only demonstrate warming in the last two years (2014-16). From 2007-14 there is no consistency in the data. They go up and down, with overall change of about 0 °C with respect to conditions in 2007-09. How does “permafrost warming reflect the Arctic and high elevation amplification of 2-m air temperature increase” when there was no increase from 2007-14? Why did the ground warm between 2007-14 if the atmosphere was not doing so? I am sure there is a simple explanation but I cannot find it in the paper, and I expect it requires more data than are presented in this paper. There are suggestions in lines 116-120, but no definitive reason is provided. I expect the authors are looking in the wrong direction, and should think about conditions below the point of observation. If the warming is amplified in the regions of interest, it is particularly awkward that no increase in temperature occurred, except for 2014-16.

We agree and found this comment very helpful. Some of the comments below address the same issue and contain valuable additional suggestions, e.g. to include snow in the analysis. We accordingly performed a comprehensive revision of our climate data analysis and explained our modifications in detail in our answer to your comment “10. P. 3, l. 114.”.

5. P. 2, l 51. “This process will further accelerate”. The values presented in this sentence are projections based upon modelling that invokes a large number of assumptions about (a) the lability of carbon in permafrost, (b) the depth distribution of the carbon, (c) the rate of active layer deepening. The data presented are for 80 years and 280 years away. I do not doubt the integrity of the scientists, but they have provided projections, and those will change as climate change progresses. To state the effects of carbon amplification with the language of quantitative certainty does not permit revision. The effects may be much greater.

Yes, we agree. This sentence could have been misleading. We reword to: “This process is projected to augment global warming”.

6. P. 2, l 66. *If all the data were available for 2007-16 I would accept this statement. But p. 5, l 220 states that data for 2007 were excluded! In fact, the initialization was for the average of data from 2008 and 2009 (l. 78). The end of the series was the average of 2015 and 2016. If the period begins in early 2009 and ends in early 2016, only 7 years of time are involved, not a decade as stated in line 66.*

This answer is partly a repetition of our answer to point 3. We assume that there is a slight misunderstanding. To calculate the rate of temperature change per year we applied linear regression on i and b using the linear model function (l_m), where $i = 2007, \dots, 2016$). Thus, the rates that describe Earth's permafrost zones in this paper are calculated from a full decade. Only the assessment of single boreholes, based on anomaly calculation explained in equations 1 and 2 has slightly higher quality criteria and is therefore based on 9 years from 2008 to 2016. This was necessary because many boreholes started temperature logging after 2007, which would be excluded in the ranking between single boreholes.

7. P. 2, l.70. *The depth of zero annual amplitude (Z^*) may be determined when the timescale of temperature fluctuation is one year. If surface temperature is changing over a longer timescale, as with these data, then Z^* is difficult to determine and will vary with the rate of long-term temperature change. This is a considerable problem, because the specific depth at which seasonal effects "are negligible" will vary with ground materials and with atmospheric trends. In practice, these considerations mean that Z^* will be overestimated. This creates another consideration, because the rate of temperature change declines with depth. It is then very difficult to compare sites, if they are being examined at a range of depths. I do not know how to circumvent this problem, except by taking an arbitrary depth (e.g. 15 m) with sufficiently frequent observations that a reliable annual mean may be calculated. The depth may need to be different in the various permafrost environments/zones discussed.*

The selection of depth is indeed a very important issue for gaining an assessment of permafrost temperature development that was discussed carefully in depth in several meetings of the Global Terrestrial Network for Permafrost. The analysis in this paper is based on their outcomes. Accordingly, permafrost depth in each is constrained by the following assumptions and quality criteria: (1) To analyze permafrost T change rates in response to atmospheric air temperature changes over multiple years, Z^* is best estimate in-between seasonal and geothermal fluctuations. To avoid depth-dependent T changes, Z^* should be kept static (at one specific depth) over the observation period. (2) Unrepeatable past and permanently installed present measurements in boreholes were based on depths and temporal resolution that vary among sites and partly within the same borehole over time. This allows numerical estimation of Z^* in only a few boreholes. A standardized (arbitrary) sensor depth does not exist but would require interpolation that again would bias the original signal. (3)

when vertical distribution of sensors and temporal resolution is high and consistent enough, Z^* should be calculated and averaged for the observation period and the closest sensor to Z^* should be selected. Where calculation is not possible/recommended, Z^* should be estimated by the PI of the individual borehole. (4) Permafrost temperature change should then be calculated from the nearest sensor measurement at or next to Z^* . The baseline of this concept and the reference period in our dataset is explained also in the Thermal State of Permafrost papers in PPP 2010). We agree that the rate of temperature change declines with depth in most cases, but it depends also on the observed time scale and the change direction. Our targeted signal, the inter-annual variation still becomes most distinct when seasonal “noise” is close to zero at Z^* . We believe that this is the best possible method bringing together both, conceptual scientific needs and methodological feasibility.

8. P. 2 I. 75. *The authors state that permafrost thaw occurred at five boreholes. The significance of the statement is not clear. Did the active layer only deepen in five boreholes out of 124? If the active layer deepened in the majority of boreholes, then permafrost thaw occurred in the majority. Does this statement mean, instead, that permafrost was eradicated in five boreholes?*

Yes, we agree that our statement needs better explanation. We reword to: “Ongoing warming above 0° C and associated permafrost thaw surpassed the depth of 10 m in five boreholes during the observation period.”. The other boreholes remained below 0°C at the observed depths.

9. P. 2, I. 93. *What constitutes the eastern Arctic and what forms the western Arctic? These are relative terms and for a circumpolar phenomenon, they vary with location of the observer. Eurocentric views might have been the norm two decades ago, but now, the frames of references are much expanded, especially considering the number of scientists – readers of the Nature journals - in China.*

We agree and have changed the terms “eastern” and “western” Arctic accordingly to North Asia and North America in this section and elsewhere (method section “Spatial de-clustering and weighted averaging”).

10. P. 3, I. 114. *The lack of correlation on an annual time scale is not surprising, given comment 4 and the data displayed in Fig. 3. There is some merit to point 1 (line 116), but the principal problem is that from 2007-14 the reanalysis data do not show any trend. Point 2 (line 118) implies, in this context, that there has been a widespread increase in snow accumulation. There must be some data available from the North American Arctic to demonstrate this, but I have not seen it in my field area (Canada’s western Arctic coast and Yukon). Point 3 is also interesting, but fundamentally, we would expect various rates of change in ground temperature in response to a change in air temperature from these factors.*

The difficulty is that Fig. 3a shows increases in ground temperature from 2008-14, when no air temperature increase took place.

Yes, causal relationships at the permafrost-air interface are part of a complex system, with a broad range of influencing variables. This comment was indeed necessary and thus very valuable. You motivated us to modify our analysis on climate variables, show more results (figures) in the manuscript and add a new section on snow to Methods. We believe that the revised version now clearly depicts ground-atmosphere relationships by including snow in the analysis. We performed the same area-weighting on the snow depth fitted to borehole sites taken from the data set of the Canadian Meteorological Centre (CMC). We found that the results explain a big portion of the lack of air T increase, especially in the lower northern regions where the constant permafrost T rise badly fits with the absent air T trend. We also included more data, i.e. longer time-series of air T data from ERA Interim to base air temperature change values as proper anomalies on the 1981-2010 reference mean. According to the WMO and BAMS this baseline replaces the 1961-1990 baseline used previously for computing temperature variations (See "Notes for Editors" at the bottom of the page in this recent press release by WMO:

<https://public.wmo.int/en/media/press-release/wmo-confirms-2017-among-three-warmest-years-record>).

Additionally, we found a more detailed assessment of the time associated to T propagation in permafrost towards our mean observed depth of 15 m in a very good paper from Lachenbruch and Marshall 1986. These authors find in a simplified model, that it can take 4 years to fully use up the energy transmitted downward at the air-ground interface. Consequently, we applied a four-year end-point running averaging on the air T anomalies and showed this data on top of the air T original anomalies. Hence this curve is a better approach to visualize the smoothed forcing of atmospheric temperature changes to the ground at our observed mean depth.

11. P. 3, l. 135. Warming of permafrost in high mountains and in the High Arctic would reflect atmospheric warming well if there had been warming in 2007-14. Instead some other explanation is required. There are many places in the discontinuous permafrost zone that respond rapidly to climate warming, but they are mostly in bedrock. Relatively few boreholes have been drilled in bedrock outcrops in continental discontinuous permafrost that are analysed by scientists, because these require industrial budgets.

The first part of this comment is related to the previous comment and addressed fully in our detailed answer above, and by the revised analysis performed. We value the comment on the sparse availability of boreholes in bedrock, and would like to emphasize that a centralized governmental funding opportunity and a joining of meteorological observatories with scientifically monitored permafrost boreholes in the

remote high latitudes and altitudes would facilitate greatly a reliable understanding of the effects global air temperature changes causing to the permafrost carbon feedback on global scale.

12. P. 4, l. 149, p. 159. The term “deep permafrost” needs considerable further thought. Lines 159 ff suggest that for these authors “deep permafrost” is a few m thick, because this is where most permafrost carbon is found. This is inappropriate. In Russia and northern North America, permafrost may be well over 400 m thick. We consider deep permafrost to be ground several 100s of m below the surface. Permafrost in the upper few m of the ground is by no means deep. Even the 30 m or more of yedoma sediments found in Alaska and Russia would not be considered “deep” by most scientists and engineers who work in the Arctic.

We agree fully that this was an inappropriate wording. Just for explanation, we were motivated to call our observations “deep” because other large time series on soil temperature data are from much shallower boreholes with a max depth of 3 m. We provide boxplots for both, sensor depth and ZAA depth distribution in the revised version (Fig. 6). We rephrased according to the median depth distribution of 15 m.

13. P. 5, l. 211. Shallow boreholes, according to the text are less than 10 m deep, but deep permafrost may not be. Or is the carbon referred to in line 149 more than 10 m below the surface?

This comment is also addressed in our answer to comment 12. The carbon we refer to is the carbon in the sediments to the median depth of 15 m.

14. P. 4, l. 214. It would be useful to know how much of the permafrost zone has been excluded by the decision to set the minimum depth of temperature record to 10 m. Does this criterion, which means that few data are considered from permafrost thinner than 10 m, lead to 25% of the permafrost zone being unrepresented in the statistics, or is the proportion of the area different?

We refrained from including this number in the main text, because it represents a) the availability of data, and b) the distribution of the depths until which boreholes have been drilled and thermistor chains have been installed, but does not contribute to understanding of the thermal permafrost system. The reason why we excluded shallow boreholes is that they are clearly influenced by short-term (seasonal) weather perturbations and caused a bias to our decadal permafrost T change rate at Z* in previous tests. However, to provide this information of the availability of shallower permafrost monitoring sites to the modelling community we now included the proportions of the number of boreholes that have met the GTN-P quality criteria but have been excluded only due to their shallowness below 10 m. We stated in the manuscript: “Boreholes that fulfilled the GTN-P quality criteria but were not included in

this analysis due to depth constraints, represented 22.6 % of the original data set. 8.6 % were excluded from the Arctic continuous data set, 23.4 % from the Arctic discontinuous data set, 30.0 % from the mountain data set and 57.1 % from the Antarctic data set.”.

Figures

Fig. 1. There are 17 locations in part a and 14 lines recognizable in b. The colours indicate that the ground temperatures in the Swiss Alps are comparable to the northern coast of Greenland/Ellesmere Island.

Yes we agree. There are 17 boreholes in both parts of the figures. In order to improve the readability the borehole IDs were included into part b and the colors have been slightly adapted and sorted according to temperature not to geographical distribution.

Fig. 2. The coldest permafrost in map a is in the mountains of Alberta. This is a surprising result, not well known before. Is permafrost there thicker than on Ellesmere Island?

Thank you for finding this mistake. It was a bug during processing, this borehole was filled with same data as Bull Pass in Antarctica. We deleted it from the analysis and checked again the other boreholes but found no further mistakes.

Fig. 3 has been considered at length in this review.

Yes. Accordingly, we split up this figure to show permafrost temperature changes per zone in more detail, and thus separately from air T changes. We now summarize potential climate impacts including snow in the new figure 4.

Fig. 5. The mean temperature within the active layer characteristically decreases with depth due to seasonal change in thermal properties (the thermal offset). Why has this not been shown on the diagram? Does this diagram imply that most boreholes under consideration are in bedrock or dry ground? The thermal offset is prevalent in ground with high water content, where the “latent heat effect” (line 138) is important. The unfrozen ground below permafrost is not commonly considered to be talik, unless, of course, there is frozen ground below this layer.

Yes, we agree. Given the fact that most scientifically monitored boreholes are in sediments we changed the figure accordingly by adjusting the MAGT dashed line to represent also the zero curtain effect. We also replaced “Talik” with “Non-cryotic”, to make it valid for all kinds of subground material.

Fig. 6. It is confusing to indicate discontinuous permafrost in the same colour as the boreholes in the continuous zone for North America.

Yes, we agree. We changed the colour accordingly.

Reviewer #2 and answers:

This manuscript describes a collection of boreholes for permafrost temperature monitoring. The authors describe the network and provide initial results on recent temperatures and temperature trends. The analysis is straightforward, for the most part, and includes a nice discussion on spatial de-clustering and measurement error. Overall, I found this paper suitable for publication. This dataset will provide a valuable constraint for models. I have a few minor suggestions for improvement.

We thank the reviewer for the time and effort spent in reviewing the manuscript. We have addressed all points raised in the revised version of the manuscript. Here, we respond to the comments and outline the associated modifications.

1. I would change “will” to “could” on line 52 since the modeling of the permafrost climate-carbon feedback remains poorly constrained and models do not show strong agreement.

We agree. In accord to Reviewer #1 similar comment we changed “will” to “was projected to”.

2. I would say that permafrost is not yet adequately integrated into MOST ESMs. Several groups have been working to incorporate permafrost processes into their models. I would also say that the lack of permafrost temperature observations is just one of the reasons why permafrost processes have not been incorporated. Another reason, for example, is that the deep soils needed for permafrost modeling require long (and costly) spinup times. In general, from my perspective, the advantage of having observed permafrost temperatures and their trends is that they provide a way to evaluate the skill of models in terms of their ability to reproduce the permafrost temperatures and their trends.

We agree and reworded to, “Despite this, permafrost change is not yet adequately incorporated into most of the Earth System Models¹³ that are used for IPCC projections for decision-makers. A major reason is the absence of a standardized global data set of permafrost temperature observations for model validation.”.

3. Reading this through raised a question for me. Are there borehole measurements outside of the permafrost domain? If so, it might be worth at least a brief discussion of these datasets here. To provide a good test of an ESM, it would be preferable to examine trends within and

outside of the permafrost domain, which would allow researchers to assess the uniqueness (or not) of the permafrost zone with respect to soil temperature trends.

This is indeed an interesting idea which would surely be worthy of an attempt for conducting analysis in a follow-up study that could be done by other authors using the new data set. One hurdle, however, is that the depth of zero annual amplitude in non-permafrost ground (Z^* , not restricted to permafrost settings) exceeds the WMO recommendations for depths measured at meteorological stations (maximum 3.2 m, e.g. Roshydromet). Hence, there is no data set existing, which we could compare to the Z^* depths in our permafrost T data set. A second problem is that groundwater flows in non-permafrost soils mean that vertical conductive heat flow may not be the primary determinant of ground temperatures.

4. The last sentence (or two) in the methods section seems important. "Enhancing existing monitoring sites through co-location with meteorological stations would also facilitate improved understanding of microclimate and buffer layer influences and also provide the data necessary for a full assessment of permafrost responses to ongoing climate change." Perhaps this sentence or something akin to it should be incorporated into the main text.

We agree very much. With this sentence we appeal to policy makers to see the importance of permafrost observation at meteorological station with governmental funding. We removed the (slightly modified) sentence from the method section and added it in the final paragraph of the main text.

Reviewer #3 (Breck Bowden) and answers:

This paper reports a synthesis of an important, global data set of mean permafrost temperatures that will of interest to a wide audience of researchers, managers, and policy-makers. The synthesis is original and likely to influence thinking in the field of climate science, especially in polar regions. However, there are a number of unclear and inaccurate statements that need to be addressed before this paper is acceptable for publication in Nature Communications.

In my comments below "L#" refers to the line number(s) in the paper. I have attached an annotated copy of the PDF which the authors might find useful.

We thank Dr. Bowden for the time and effort spent on his detailed review. We are particularly grateful for the comments and suggestions on how to improve our statistical analyses with respect to the p-value data matrix. We addressed these

points thoroughly in the revised version of the manuscript and especially revised our R code by adding the Wilcoxon Signed Rank test and a False Discovery Rate adjustment to account for non-normally distributions and false positives. We also value the detailed edits in the PDF version of the manuscript which helped to improve the readability and precision of the text.

The most important points requiring clarification are:

L30-31: The global mean permafrost value reported in the abstract appears to be an arithmetic mean. I think a mean weighted by the areas of the defined regions would be more appropriate. Are these means different?

This is a valid comment. We have now added an area weighting to our methodology by accounting for the areas of each of the 10 permafrost zones and added them to our weighting by indexes. Indexes have been developed to reduce the influence of seasonal fluctuations from above and the geothermal influences from below. We now additionally calculated the areas (square kilometers) by plotting the boreholes in the IPA permafrost maps. For Antarctica we used area values from the literature. We thus reworded the weighting method to “Area-weighting, spatial de-clustering and indexing” and described it in detail in our methods. The revised weighting impacted the change rates per zones and globally. A good side effect is, that the low number of boreholes in Antarctica does not impact the global value too much anymore, due to the large area difference (e.g. Antarctica is just 0.5 % of the total area in the Arctic continuous permafrost zone). Overall the values in the Arctic decrease slightly to 0.39 ± 0.15 °C and 0.20 ± 0.10 °C dec⁻¹ in the continuous and discontinuous permafrost zones, respectively. The general strong temperature warming trend is still distinct. Mountain permafrost was not influenced enough by area-weighting to change the values given the number of digits used. Antarctica is treated as one zone and hence area-weighting was not applicable. The global mean was also area-weighted, resulting in 0.29 ± 0.12 °C dec⁻¹.

*L154-155: According to Fig. 2c, it seems that there are some borehole sites at which cooling has occurred. Although uncommon, this does not support the statement that the “entire zone” of continuous permafrost in the Arctic is warming. Furthermore, across all of the boreholes, a substantial portion ($[124-71]/124 * 100 = 43\%$) either cooled (12) or did not warm or cool ($124-71-12 = 41$). This is mentioned briefly at lines 74-75. I think a more balanced statement is warranted in the abstract. For example, “While cooling was observed in some boreholes and others did not demonstrate significant warming over the decade, the overall mean temperatures in all 10 regions increased significantly.”*

Yes, we agree. We reworded accordingly to: “... strong warming within the Arctic continuous permafrost zone...” in the text. It is however difficult to include the

suggested statement in length in the abstract because we do not fully describe the method in the abstract, but mentioning of the 10 zones would require explanation. We therefore do not state that “entire” regions are warming but stick to averages. If possible we would like to keep it concise to enable for more people to understand the core message fast.

L220: I understand that it is common practice in papers on climate change to refer to temperature "anomalies". These are usually understood to mean departures from a long-term and well-established norm and are interpreted to be unusual. I fully understand that the authors think the departures that they report here are unusual, and I agree. However, in this case, the departure is from an arbitrary, two-year mean. I think it is more accurate to refer to these as "departures" rather than "anomalies".

We agree. We reworded accordingly from “anomaly” to “departure” or simply “temperature change” or “ $\Delta\bar{T}$ ”, adjusted to the readability of the individual section. We included the established reference period 1981-2010 in our revision of air versus permafrost temperature comparison. We continue to use the term “anomaly” for air temperature changes. This also facilitates to distinguish intuitively between permafrost and air T changes for fast readers.

L246-247: How many monthly measurements were required from a mean T value to be valid? Was it necessary for there to be at least one value in each of 12 months? If not, why not? See also the comment at lines 250-253.

The quality criterium for a valid mean annual ground temperature in a borehole is described in detail in the manuscript. The depth we observe is distributed closely at the depth of zero annual amplitude Z^* , slightly above, or below, depending on sensor and data availability. If data is only available from above the Z^* , mean calculation is only allowed to be included in our analyses if every month is represented in the measurements. If the data indicates Z^* or below Z^* , a lower resolution is allowed, down to one measurement per year. At these depths, seasonality is negligible. This rule is irrevocably necessary because a high number of boreholes at remote sites are monitored by expedition teams only once in the year. A calibrated thermistor chain is lowered in the borehole until equilibrium is reached and the T value is retrieved manually. Even though these are logistical needs, the method is scientifically correct at greater or equal to Z^* and a multi-year observation period. The measurement period is indicated in the data set.

L247-249: The sensor selection criteria noted in lines 203-204 is important here. At how many sites were monthly data missing for boreholes with sensors below Z^ ? Am I correct that <no> borehole data were included for sensors that were shallower than Z^* and had missing monthly data?*

Monthly data are missing at 29 of 156 boreholes (=18.6%). These data are restricted to depths where lower-than-monthly measurements are scientifically legitimate. Yes, the statement is correct: Above Z^* all data included in our analyses presented in this manuscript (or elsewhere) are based on at least monthly resolution, which means to be precise - measurements were conducted once or more often in every month of the year involved. Please see detailed explanation above (to L246-247).

Lines 250-253: It appears from these statements that the data reported here are based on temperature measurements made over different numbers of months depending on the local meteorological conditions (i.e., latitude and elevation). Specifically, it appears that the permafrost means are not annual means, but a mean based on a variable portion of the year, depending on the site. Thus, the mean permafrost temperature at sites could be influenced by different seasonal characteristics. If the primary focus of this paper is on temperature change or departures from a reference, why not define that reference to be as consistent as possible? For example, select the three months that are most indicative of boreal or austral "winter" conditions?

We are considering overall trends of mean annual ground temperatures over a decade. 116 of 127 (91.3%) boreholes with monthly or greater resolution of measurements are based on calendar mean calculation from January till December. Different periods are in 63.6 % based on December till November succession. This means that the bulk data is comparable to calendar years. Logistical challenges did not allow a 100% calendar year data set. However, the observation in this manuscript is basically at or near to Z^* in the overall analysis and hence intra-annual fluctuation minimized. Exclusion of the few meteorological years even though very close to Z^* would result in loss of spatial resolution and therefore reduce the representation of the presented regional and global mean. It is further not possible to define three month, because many of the boreholes are visited and measured only once per year (below Z^* of course).

L284-291: It is not clear whether the numbers preceding each weighting criteria are in fact the weights or are simply index numbers. This should be clarified. How were the weights actually used in this assessment?

Thank you very much for this valuable comment. Yes, we used the borehole criteria as indexes for weighting the mean calculation. Highest quality boreholes with best data representation have been multiplied according to the index before calculating means. As also described in response to (and thanks to) your comment above for L30-31, we now included area weighting based on the surface areas of the 10 world zones in square kilometers. We described the weighting method in detail in the methods section in the revised manuscript.

L329-332: It is unclear to me what values have been assessed in the matrix of t-tests provided as supplementary data. I presume these are the permafrost anomalies, but this should be stated explicitly. If the data are not normally distributed, why use a t-test? A non-parametric test (i.e. Wilcoxon Signed-Rank test) might be more appropriate. Furthermore, there are 100 individual tests in this table. Simple probably suggests that perhaps 20% of these would be “false positives.” Thus, in assessing this matrix of results, some sort of Bonferroni-type adjustment should be included. If this supplemental data table is to be included with the final manuscript, the p values should be reduced to no more than 3 significant digits. There is little meaning to a p value with 7 significant digits.

We agree and performed a Wilcoxon Signed-Rank test to calculate p values. We found that False Discovery Weight (FDR) is even better than Bonferroni type adjustment in the case of our larger dataset. Bonferroni pays more attention to keep type I errors low and hence excludes any potential false positives, but also creates a very high number of false negatives (type II error). Thus, it reduces the false positives at the expense of false negatives. FDR is a bit less strict in excluding false positives, but therefore generates not as many false negatives, but still a high number (about 30 percent) in large datasets.

In our revised analysis we calculate p-values of weighted permafrost anomalies from each zone and each year, reduced to 3 digits.

We did not aim to publish the matrix of p-values along with the manuscript, because testing the significance of data is in sympathy with discussion on the significance of the test itself (e.g. <https://www.nature.com/news/scientific-method-statistical-errors-1.14700>) and whether or not it is relevant for the design of a study. We claim that our main outcome, represented by indexed area-weighted means, represent the prevalent trend on temperature change in permafrost in course of climate change. As we described in the text, however, climate and ground temperature is – due to the complexity of the Earth system – spatially not homogenous and thus, p-values vary. We explain both, the original Wilcoxon Signed-Rank computed p-values and the FDR adjusted p-values in the data quality assessment chapter.

L322-323: Why arbitrarily pick a mean accuracy that you know is lower than the accuracy of some sensors in the network? It seems that the accuracy should be set nearer the least accurate sensor in the network. If the accuracy of 0.25°C is anomalously high, that should be stated. Or some other criteria other than “arbitrary” should be given to justify this important threshold.

Yes, this is a valid comment. Measurement accuracy in boreholes is not very easy to describe due to the variety of techniques used. We therefore re-wrote the section on measurement accuracy completely and added the necessary explanations. 0.25 is just a very high absolute precision of only few boreholes. This error is important in the

case you want to compare the absolute “static” temperature of two boreholes with each other. However, expected measurement errors in changes over time within the same borehole are much lower, allowing us to present a sound temperature change estimate. Please find details in the new extended accuracy chapter.

The following are a number of edits I suggest to the authors to help clarify other aspects of the paper:

L58: Need a better transition here: "which" rather than "these"?

We agree and revised accordingly.

L71-73: The rates are not "decadal"; rather, they are "rates calculated over a decade".

We agree and revised accordingly.

L105: Were these really "analogue" calculations? The means appear to have been calculated using digital data.

We agree and revised accordingly to “same calculation method”.

L122: "de-noising" is an awkward term. Rather, "acts as a filter that removes rapid variation caused by..."? Is this variation really noise (i.e., error) or real variation that obscures the long-term trends the authors want to reveal?

We agree and revised by using the suggested alternative wording.

L126: "rates of change"

We agree and revised accordingly.

L138: The reference here is unclear. Did air temperatures increase 5x faster in the discontinuous zone than air temperatures in the continuous zone? Or did the discontinuous permafrost temperatures increase 5x faster than the permafrost temperatures? Specifically, it is unclear if the final reference to "permafrost temperatures" refers to the permafrost itself or to the air temperatures in that region.

Thank you for this valid comment. According to the new results gained by the additional analyses performed as suggested by Reviewer#1 we deleted this section.

L186-187: Is there a place, either on the GTN-P website or in the supplemental data, where key aspects of the types of harmonization, filtration, and standardization employed to make the data set coherent could be recorded? This might be important for future users of this global data set.

Yes, there is a detailed description on our data management system available online at www.gtnp.org, if you click on “data”. You will find the requested key aspects there, and for the applied filters to create the data set used here, also in the method section in the manuscript. However, we did not record every single format standardization step we did to individual data sheets. Basically we did not change original data, but performed quality check, aggregation and visualization. You can find the original data online at <http://www.gtnpdatabase.org/boreholes>.

L203-204: How many sites had sensors above Z^ vs below Z^* ?*

We calculated that of 92 boreholes with reliable Z^* indication 18 measurements are from above, 55 directly at and 19 below. This means: 19.5% of measurements are from slightly above Z^* , 59.8% are representing Z^* and 20.7% are below Z^* . The rest of boreholes that have no reliable indication of Z^* contribute with data from 17.1 m depth on average, which is well below the average of all indicated Z^* values (14 m) and therefore we assume that our results are reliable. We accordingly added this to the method section:

“19.5% of measurements are from above Z^* . 59.8% of measurements are representing Z^* and 20.7% are from below Z^* . Measurements from boreholes that have no reliable indication of Z^* contributes with data from 17.1 m depth on average, which is well below the average of all indicated Z^* values (14 m).”

L208-209: The phrase "estimation of the scientist responsible for the borehole" is vague. What criteria were used to ensure consistency at a site, over time and across the network?

Thank you very much for this comment, yes we agree. This was written from the perspective of the first author only. We revise accordingly to: “(2) by visual detection of summer and winter temperature measurements plotted versus depth (Fig.7).”

L215: The phrase "showed little seasonal fluctuation" is vague. I assume the seasonal fluctuations were less than the instrument precision and accuracy. If so, this should be stated.

Yes, we agree, and reworded accordingly to “... and seasonal fluctuations were less than the instrument precision and accuracy”.

L220: The reason that the 2007 borehole data were excluded should at least be mentioned briefly here. From the methods, I think this was because there were too many missing values in 2007.

Yes, this is true. The estimations for single boreholes departures is based only on boreholes with high quality criteria, including that data in involved years must exist. Therefore 2007 was excluded because IPY started in 2007 and hence many boreholes started to record full years earliest in 2008. We added the following wording: “Continuous (full-year) records started at a large number of borehole sites during the 4th International Polar Year (IPY) period in 2008. To base the reference period for the departure calculation on the largest possible number of boreholes we excluded 2007....”

L300-301: This is confusing. It appears that only two years (0 and 1) are used to calculate Mean Annual Air Temperatures? However, equation 3 suggests that the entire available record was used. This should be clarified.

Thank you for the comment, we may have not explained this clear enough. Year 0 and Year 1 are here just representing the calculation method of a mean annual value in any year, the equation is not restricted to two years. We reworded accordingly in more detail, also in accord to Reviewer #1 suggestions and in line to our additional analyses on air temperature reanalysis for comparison to our results on permafrost change.

L310: To be clear, I suggest adding "the >air< temperature".

We agree and reworded even more to “To calculate the rate of temperature change over a decade, we apply linear regression on $\hat{T}_{i,b}$ for all i and b using the linear model function (lm) in the R environment and the slope of the linear regression in an annual array between 2004 and 2016 and multiplied the annual change rates by ten”.

L337-347: This assessment of the data seems better placed after line 163, just before the methods section.

We agree and shifted this (slightly modified) section to the end of the manuscript final text.

L507 (Fig. 1): It is difficult to make out the color gradations for each borehole on the figure and then transfer this to the map. It would be helpful to have the borehole numbers on the figure.

Yes, we agree. In order to improve the readability of the figure, the borehole IDs were included in part b and the colors have been adapted.

L533-534 (Fig. 3): The temperature confidence intervals as dashed lines are difficult to decipher. The color difference between the Arctic Discontinuous and Mountain data is especially difficult to make out.

We agree and split up the graph with new area-weighted data in four separate plots.

L540 (Fig. 4): What are the error bars? SD? SE? Range? Inter-quartiles?

We added to the caption of the figure and in the methods: “Boxplots represent 25-75% quartiles and whiskers are 1.5 interquartile ranges from the median. The median is shown as a black line.”

Additional note to your comment in the PDF, which we did not find in your list of comments above:

L21-22: I find these statement problematic given that the GTN-P exists and has posted permafrost temperature data files and maps. However, this does not obviate the importance of this paper, which is to <synthesize> the recent results from this important network. This high-level synthesis is an important contribution to the literature. I suggest that these sentences could be deleted without materially changing the manuscript.

Thank you very much. The GTN-P published only the distribution of boreholes, metadata statistics and in some talks the state of permafrost temperature, but not the development over time. This is not existing. This paper is the first and only paper developing a comprehensive numerical assessment of the global permafrost temperature development. We reworded to “Yet to date, no globally consistent assessment of permafrost temperature change has been compiled”.

Reviewers' comments:

Reviewer #1 (Remarks to the Author):

I have now only a few comments as many of my previous remarks have been addressed.

The paper needs to be very carefully edited to bring the English writing to the standard anticipated by readers of the Nature journals. I believe the person who does this job needs to pay very careful attention to speculation in the paper that seems to have been inserted as a sales-pitch rather than the on the basis of the results. The last sentence of the abstract is a case in point.

I do not like the rate of warming in units of per decade, because there is only one decade of data. I would prefer the warming to be described as a total because that is what has been measured.

The definition of permafrost in line 43 is unusually loose and is not standard. "Multiple years" are undefined. How many years constitute "multiple". 10, 20, 30?

There is a significant mismatch in scale between the data for individual boreholes and the climate reanalysis. This is probably most serious for snow cover. The data on snow cover are problematic, because it seems that the authors use snow depth, but do not present dates for arrival of snow or when maximum snow depth occurs. Snow conditions vary significantly over short distances, and so I would treat the mismatch in scale between the climate data for snow and the response of the individual boreholes much more cautiously than the paper does at present. I am not sure the data presented warrant a firm conclusion that snow depth is the variable that accounts for the increase in ground temperature in the discontinuous permafrost zone.

I mentioned previously that changes in ground temperatures at at 10 m will likely have faster response to climate changes than at 20 m. I think this point, while not lost on the authors, needs to be explicitly excused through their consideration of the depth of zero annual amplitude, i.e. they might beneficially point out such differences. I realize their data set is variable on this point.

The comments in lines 154-166 seem mysterious to me. I do not understand what changes that terminated more than 300 years ago have to do with the upper 20 m of the ground. My original point on this topic concerned the last 40 years, not 400!

Reviewer #3 (Remarks to the Author):

I have read the responses by the authors to my comments as well as their responses to the other two reviewers' comments. I think the authors have made substantial changes to the manuscript that have improved an already worthy paper. I do think there are a few items that require further clarification prior to publication. However, I would not expect these clarifications to hinder acceptance of this manuscript. In the interest of time, I would be happy for the Editor to review responses by the authors to the following queries and suggestions and make a decision on publication without seeking my further opinion. Of course, I'd be happy to review these responses if need be.

My most substantial comments are as follows:

L155-166: I think the first sentence in this set of lines (regarding permafrost "memory") is useful. But I'm confused about the remaining lines. Why mention a potential rebound from the Little Ice Age - which is largely speculative I think - only to dismiss it as "not expected" to have any substantial effect? In the final sentence, the statement is that the differences between "colder years" relative "warmer years" were insufficient to explain warming permafrost. I presume the reference to "cold/warm" is in reference to SAT? If so, that should be made explicit. Frankly, I would recommend to delete all of these lines (with the possible exception of the first sentence, L155-159). The strongest evidence you have is the snow data, which I find sufficiently compelling.

L274: If you immediately exclude the 2007 data then why define this set as starting with 2007 data?
L282-284 and Eqns 1 and 2. It is not clear to me why you need both of these equations. When Eqn. 1 is subsequently regressed on years, then the slope b is the rate of change, as you note later. (But see my next comment.) You could simply solve the regression equation for the beginning and ending years to get a total mean estimated departure over the period, which is essentially what Eqn. 2 does.

L288-290 in reference to Eqn. 1: It is perhaps a small point, but I would not do this regression over the full range of i (years). You have used the first two years (2008 & 2009) as a reference and so regression over these first two years is guaranteed to include variance around the mean of these two values. Your real question is in regards to the rate at which <subsequent> years depart from that reference. Thus, I think it is unfair (and perhaps counterproductive) to include these two years in your regression. I would expect the intercept of the linear regression fit to the date excluding 2008 and 2009 to fall very near the actual reference value. If it does not, this might indicate a non-linearity in the temporal trend.

L290-293: If you follow my logic regarding the previous comment, then I think it is easier to accept this definition of your reference point. If in some years you include 1, 2 or 3 years in the mean and then regress over all <available> years, then you are weighting the starting point of your reference years differently in the regression. If you treat the reference mean as a parameter only (i.e., not as input data) and regress over all <other available> years then you eliminate this potential bias. Also, I thought you excluded all 2007 data from this analysis. Was 2007 data used to define reference values?

I have several simple editorial suggestions intended to shorten or clarify the narrative. The authors are welcome to accept or reject any and all of these suggestions:

L30-31: If you want to save a few words here, I would replace "Long term permafrost warming observed at single sites is also true at global scale, where temperature..." with "Globally, permafrost..."

L49: Replace "dramatic implications" with "important consequences".

L54: Delete "mostly"

L67: Delete "newly"

L76-77: Was the global mean also computed as the area-weighted mean of the regions?

L83: The implication of this statement is not clear: "surpassed the depth of 10 m". What is important about 10 m? Was this the Z^* for those boreholes? Or was this the maximum depth for those boreholes?

L121: Do you mean "time lag" here instead of "time scale"?

L167-171: I suggest to start this sentence with "We attribute..." and to delete "...is attributed to..." in L168. Otherwise it sounds like an established fact that you are merely repeating.

L263: Insert "longer-term" between "reflects" and "climate change"?

Fig 6. I like this graph. But it took me a moment to realize what it was telling me. I suggest reversing the y-axis (depth) so that it reads downwards. Then it is more obvious that the median sensor depth was deeper than the media Z^* .

I like the new forms of Figures 3 and 4. Very nice.

Breck Bowden

Revision Notes

15 September 2018, Biskaborn et al.

Reviewer #1 and answers:

I have now only a few comments as many of my previous remarks have been addressed.

The paper needs to be very carefully edited to bring the English writing to the standard anticipated by readers of the Nature journals. I believe the person who does this job needs to pay very careful attention to speculation in the paper that seems to have been inserted as a sales-pitch rather than the on the basis of the results. The last sentence of the abstract is a case in point.

We are very grateful to Chris Burn for reviewing our manuscript and we agree with all of his comments. His comments were helpful in improving the manuscript. Revisions have been made, including professional (external) English proof reading, to remove speculation and to ensure that statements are based on results (i.e. removed “sales-pitch”).

I do not like the rate of warming in units of per decade, because there is only one decade of data. I would prefer the warming to be described as a total because that is what has been measured.

We agree. This is indeed a valid point. Our results do represent only one decade. Accordingly, we changed the unit in the abstract to °C and changed the unit in the main text to °C dec_{Ref}^{-1} . This new unit implies that we refer to a defined decade (not to any decades).

The definition of permafrost in line 43 is unusually loose and is not standard. "Multiple years" are undefined. How many years constitute "multiple". 10, 20, 30?

We agree with this criticisms and stick to the original wording presented by the IPA: “at least two consecutive years”.

There is a significant mismatch in scale between the data for individual boreholes and the

climate reanalysis. This is probably most serious for snow cover. The data on snow cover are problematic, because it seems that the authors use snow depth, but do not present dates for arrival of snow or when maximum snow depth occurs. Snow conditions vary significantly over short distances, and so I would treat the mismatch in scale between the climate data for snow and the response of the individual boreholes much more cautiously than the paper does at present. I am not sure the data presented warrant a firm conclusion that snow depth is the variable that accounts for the increase in ground temperature in the discontinuous permafrost zone.

Thank you very much for the careful consideration of the air-snow-permafrost relationships presented in our manuscript. We agree that the timing of snow is a very important variable needed to validate the warming effect of insulation caused by snow. We performed additional analyses to include the timing of snow, including three variables, shown in figure 4: snow onset, maximum insulation effect of snow, and the end of the snow melt. The trends in all three variables support our findings. We carefully described our additional analysis in the method section and discussed the results in the main text.

Yes, we are aware about the mismatch between permafrost temperature observations in boreholes and the climate data from reanalysis data sets. The scale mismatch is a significant issue for snow because there can be substantial difference between snow cover over very short distances. We therefore conceived the snow analysis in a conservative strategy: We assumed that the onset of snow will be applicable for an entire grid cell, and hence the borehole, as soon as the reanalysis value is 6 cm for at least 6 days in a row (following usual criteria in modelling). Additionally, we assume, that permafrost at borehole sites near the zero annual amplitude is a three-dimensional system being influenced by a large area, represented in the reanalysis data. This is the best we can do without having meteorological observation data available at boreholes. The data analysed in our manuscript still clearly indicate that permafrost is warming in course of recent climate change. Snow in the Arctic, and especially the discontinuous zone, has a distinct effect that facilitates this warming trend.

I mentioned previously that changes in ground temperatures at at 10 m will likely have faster response to climate changes than at 20 m. I think this point, while not lost on the authors, needs to be explicitly excused through their consideration of the depth of zero annual amplitude, i.e. they might beneficially point out such differences. I realize their data set is variable on this point.

Yes, we agree that this point needs more clarification. As we showed in Fig. 6 the observed depths in the boreholes vary around the mean zero annual amplitude Z^* . The figure also shows that Z^* varies, which is why it is not possible to keep the observation depth constant while accounting for a proximity to Z^* allowing

assumptions on climate changes beyond weather phenomena. We already analyzed and reduced substantially our previously much larger dataset that included also shallower boreholes in the long preparation phase by the GTN-P community for this manuscript. As requested by the other Reviewers before we included details about this process in the method section:

“Boreholes that fulfilled the GTN-P quality criteria but were not included in this analysis due to depth constraints, represented 22.6 % of the original data set. 8.6 % were excluded from the Arctic continuous data set; 23.4 % from the Arctic discontinuous data set; 30.0 % from the mountain data set; and 57.1 % from the Antarctic data set.”

We now added a new sentence:

“Similar temperature trends of the remaining shallow (≤ 12 m) and deeper (> 12 m) boreholes in the data set confirm that the observed depths near Z^* (Fig. 6b) provide a representative sample tracking climate variability coherently.”

In other words, after excluding the shallow boreholes that showed indeed a too strong short-term climate influence, the data set shows a trustworthy temperature change spreading among depths. To make this as clear as possible, we included a second graph (b) in Fig. 6 showing the similar temperature distribution at depths shallower and deeper than the median Z^* in the data set.

The comments in lines 154-166 seem mysterious to me. I do not understand what changes that terminated more than 300 years ago have to do with the upper 20 m of the ground. My original point on this topic concerned the last 40 years, not 400!

We agree. This section was misleading. We changed to:

“Another important factor that explains the general discrepancy between annual changes in permafrost and the atmosphere is that permafrost progressively with depth “remembers” the surface temperature history of past several decades^{24,36}. The temporal dimension of episodes with lower air temperatures between 2009 and 2013 in the Arctic (Fig. 4a-b), and around 2012 in the mountains (Fig. 4c), relative to preceding period of higher air temperatures, however, was not large enough to sustainably impact the general warming trend of permafrost. We can also exclude climate events from hundreds of years ago, e.g. the Little Ice Age, as an influence on temperatures at the depths near Z^* monitored in our study.”

Reviewer #3 and answers:

I have read the responses by the authors to my comments as well as their responses to the other two reviewers' comments. I think the authors have made substantial changes to the manuscript that have improved an already worthy paper. I do think there are a few items that

require further clarification prior to publication. However, I would not expect these clarifications to hinder acceptance of this manuscript. In the interest of time, I would be happy for the Editor to review responses by the authors to the following queries and suggestions and make a decision on publication without seeking my further opinion. Of course, I'd be happy to review these responses if need be.

We are very grateful to Breck Bowden for reviewing our manuscript again. Your comments helped very much, especially the comments on the statistical and math part of the manuscript, i.e. the area-weighting. We carefully considered and acknowledged all your new comments as well. Thank you very much. This is most appreciated by all authors.

My most substantial comments are as follows:

L155-166: I think the first sentence in this set of lines (regarding permafrost "memory") is useful. But I'm confused about the remaining lines. Why mention a potential rebound from the Little Ice Age - which is largely speculative I think - only to dismiss it as "not expected" to have any substantial effect? In the final sentence, the statement is that the differences between "colder years" relative "warmer years" were insufficient to explain warming permafrost. I presume the reference to "cold/warm" is in reference to SAT? If so, that should be made explicit. Frankly, I would recommend to delete all of these lines (with the possible exception of the first sentence, L155-159). The strongest evidence you have is the snow data, which I find sufficiently compelling.

We agree. This section was misleading and we deleted most of it or revised accordingly. Reviewer #1 also criticized the involvement of LIA. However, he also stated that snow depth alone is not sufficient enough to explain permafrost warming. Furthermore, the models we mention in the text have shown that permafrost in terms of thermodynamics is complex and responsive but also a very slow system. Snow indeed can have a warming effect in early-mid winter but also a cooling effect if it prevails in latest winter. However, the mean annual permafrost data we have cannot account for intra annual fluctuations but assesses primarily the inter annual variability which is largely driven by long term climate fluctuations. We would therefore prefer to keep a drastically shortened and corrected version of the section that explains the lag of permafrost temperatures behind air temperatures.

L274: If you immediately exclude the 2007 data then why define this set as starting with 2007 data?

We exclude 2007 data only to get a higher quality of the calculation of T differences for every year and to compare boreholes between each other. We use 2007 data to

calculate the mean temperature changes over one decade in the “world zones” to get a better spatial representativeness. To not define too many similar data sets at the start, we define one data set including all available years.

L282-284 and Eqns 1 and 2. It is not clear to me why you need both of these equations. When Eqn. 1 is subsequently regressed on years, then the slope b is the rate of change, as you note later. (But see my next comment.) You could simply solve the regression equation for the beginning and ending years to get a total mean estimated departure over the period, which is essentially what Eqn. 2 does.

We need all of the equations due to the following: Eqn. 1 calculates the T difference for every year in the observed decade to show the inter annual variability. Eqn. 2 calculates a standardized total T change between two defined mean periods to compare (to rank by highest degree of warming) boreholes between each other. To express the overall T change rate in the world zones, we perform regression on the entire time bars of mean annual ground temperatures in the boreholes between 2007 and 2016 following the specified criteria. We do not include multiple-year-means in the regression.

We are aware that the method is described in a mathematical way and that it will help to add more explanations in the text. Accordingly, we performed a careful revision of the math description and edited the section around Eqn. 1 and 2. It should be easier to follow our approach now. Thank you.

L288-290 in reference to Eqn. 1: It is perhaps a small point, but I would not do this regression over the full range of i (years). You have used the first two years (2008 & 2009) as a reference and so regression over these first two years is guaranteed to include variance around the mean of these two values. Your real question is in regards to the rate at which <subsequent> years depart from that reference. Thus, I think it is unfair (and perhaps counterproductive) to include these two years in your regression. I would expect the intercept of the linear regression fit to the date excluding 2008 and 2009 to fall very near the actual reference value. If it does not, this might indicate a non-linearity in the temporal trend.

Please see our answer above. In short: We do not perform regression on mean periods but only on pure time bars. There is indeed a mathematical intercept always between 2008 and 2009 in Eqn. 1 (Fig. 3). However, this is not a mistake. The slope of the curve still indicates correctly the relative T increase.

L290-293: If you follow my logic regarding the previous comment, then I think it is easier to accept this definition of your reference point. If in some years you include 1, 2 or 3 years in the mean and then regress over all years, then you are weighting the starting point of your reference years differently in the regression. If you treat the reference mean as a parameter only (i.e., not as input data) and regress over all <other available> years then you eliminate

this potential bias. Also, I thought you excluded all 2007 data from this analysis. Was 2007 data used to define reference values?

Your logic itself is correct, and we would revise the analysis if we would have done it this way. But there is probably a slight misunderstanding. Please see our answer above. We do not perform regression on mean periods but only on pure time bars in which all values are mean annual values. We excluded 2007 data only for Eqn 1 and 2, but we included 2007 data the regression analysis.

I have several simple editorial suggestions intended to shorten or clarify the narrative. The authors are welcome to accept or reject any and all of these suggestions:

L30-31: If you want to save a few words here, I would replace "Long term permafrost warming observed at single sites is also true at global scale, where temperature..." with "Globally, permafrost..."

Thank you. We agree and reworded accordingly.

L49: Replace "dramatic implications" with "important consequences".

We agree and reworded accordingly.

L54: Delete "mostly"

We agree and reworded accordingly.

L67: Delete "newly"

We agree and reworded accordingly.

L76-77: Was the global mean also computed as the area-weighted mean of the regions?

Yes, we calculated the global value as an area-weighted mean. We added this as well in the text. Thank you for the hint.

L83: The implication of this statement is not clear: "surpassed the depth of 10 m". What is important about 10 m? Was this the Z for those boreholes? Or was this the maximum depth for those boreholes?*

Originally, we just stated "permafrost thawed", because these boreholes time series develop into positive temperature values. This was criticized and hence we stated more correct that thawing surpassed 10 m – which was the observation depth (depth

of sensor). We now reworded to make this easy: "Ongoing warming above 0° C and associated permafrost thaw surpassed the observation depth of 10 m in five boreholes."

L121: Do you mean "time lag" here instead of "time scale"?

Thank you for the hint. In this case we would prefer to use a third option "annual resolution", because the time lag calculated is not only one year but 4 years.

L167-171: I suggest to start this sentence with "We attribute..." and to delete "...is attributed to..." in L168. Otherwise it sounds like an established fact that you are merely repeating.

We agree and revised accordingly.

L263: Insert "longer-term" between "reflects" and "climate change"?

We agree and revised accordingly.

Fig 6. I like this graph. But it took me a moment to realize what it was telling me. I suggest reversing the y-axis (depth) so that it reads downwards. Then it is more obvious that the median sensor depth was deeper than the media Z.*

We agree and revised accordingly. We also added the temperature distribution > and < 12 m in response to the comments of Reviewer #1 in this graph.

I like the new forms of Figures 3 and 4. Very nice.

Breck Bowden

Thank you very much, your review helped substantially to improve the quality of this study. It is very much appreciated.

Boris Biskaborn et al.

General comment:

The part of air temperature and snow data in the manuscript increased substantially during the revision process and accordingly we shifted Heidrun Matthes (being responsible for meteorological data) from the alphabetical order into the first listed group of principle co-authors.

Reviewers' comments:

Reviewer #3 (Remarks to the Author):

[Note: I have attached these comments as a separate file as well, which may be a more readable format.]

I appreciate the extensive revisions made by the authors. I've read the rebuttal comments and the revised paper. I conclude that revised paper is much improved. I do have a lingering concern about the analysis done with equations 1 and 2 and have identified a few other comments that think the authors should consider.

Extended comments on Equations 1 and 2

I continue to have some reservation about the uses of equations 1 and 2. Equation 1 provides the departure of the mean borehole temperature from a reference temperature for each year of the record (T_y, b). The regression of these annual values over years provides an estimate of the annual average change in temperature over the period of record. The total change in borehole temperature over the period of record can be determined by solving this regression equation for the first and last years of the record and taking the difference. Equation 2 produces a second estimate of the total difference in borehole temperature (T_b) by taking the mean difference between the first two years of the record and the last two years of the record. It is noteworthy that the reference temperature used in equation 1 is identical to the "initial" term in equation 2; i.e., it is the mean of the first two years of the record.

I don't have any particular difficulty with either of these equations and understand why and how the authors use them. I think they are somewhat redundant, but the authors can certainly use both if they wish. I have a somewhat greater concern about a potential instability that could be introduced in the regression of the values generated by equation 1, utilizing the reference temperature as it is defined (the mean of the first two years of the record). I have not been able to articulate this concern in my previous narrative comments and so I have included a simple Excel "thought experiment" that I hope illustrates my point. This file is provided separately.

In this thought experiment I have assumed arbitrarily that the initial permafrost temperature is -7°C (cell B3). I also assumed that there is some error in each annual measurement that is a combination of real and stochastic events. I limited this error to a maximum of 0.5°C (cell B4) in any given year. The choice of the actual values for these two variables is not particularly important. I then introduced a random factor that has a dynamic range of 100. The maximum allowable lower bound on this dynamic range is -100 and the maximum allowable upper bound is $+100$, but the dynamic range itself can only be 100 units. Thus, this factor is perfectly "centered" if the random lower bound (RLB, cell B5) is -50 and random upper bound (RUB, cell B6) is $+50$. In this case the Excel random number generator will on average produce as many negative numbers (between 0 and -50) as it does positive number (between 0 and $+50$). If the boundaries are skewed upward so that $\text{RLB} = -25$ then the RUB must be $+75$ (to preserve a dynamic range of 100). In this case the Excel random number generator will produce more positive numbers (between 0 and $+75$) than it does negative numbers (between 0 and -25). In use, each random number that is generated (each year) is divided by 100 to create a fraction. That fraction is multiplied by the 0.5°C maximum temperature excursion and then added to the previous year's borehole temperature. This is done for each year (cells A11 to A19) to create a synthetic borehole record (cells C11 to C19) from the random annual temperature excursions (cells B11 to B19). If the dynamic range is "centered" then the temperature record is simply random variation around the mean value of -7°C . But if the dynamic range is skewed upward, it generates a "warming" trend and if it is skewed downward it generates a "cooling" trend. The magnitude of the skew determines the "strength" of the trend. A new realization of this model can be created by clicking the Function-F9 keys together. This will re-randomize the model and quickly allows the user to see numerous iterations of the same set of conditions.

The Excel spreadsheet calculates $dT_{y,b}$ in cells D11 to D19 (highlighted in yellow) and also the reference temperature ($1/2*[T_{2008}+T_{2009}]$) in cell H10. This data is then used to generate the data in the block of cells from E12 to I16 (also highlighted in yellow). This block calculates the slope of $T_{y,b}$ by year, the total change expected over all years (9 in this case), the r^2 for this regression, the value of dT_b from Equation 2. It does this for two different treatments of the data. The first treatment (cells H13 to H15) is the treatment as described in the paper, in which the reference years are defined as in cell H10 and all years in the record are assessed against that reference. The second treatment of the data (cells I13 to I15) excludes the data for the reference years from the regression, so as not to “double count” their influence on the regression. This is the situation I attempted to describe in my previous review. The value of dT_b in the merged cell at H/I16 is the same for both treatments. But it can be compared to the alternative forms of the same metric (the total change over 9 years, the “decadal” change) for the two different treatments of the data (cells H14 and I14.)

My conclusion from playing with this model is that if the warming trend is strong (i.e., the skew of the dynamic range is say -5 to 95), then repeated re-randomizations of the model typically create a well-behaved, upward, linear trend and the various estimates of $T_{y,b}$ and T_b are reasonably similar with high r^2 values for the regression of $dT_{y,b}$ versus years.

However, if the warming trend is weak (say, a RLB = -45 and RUB = +55) then there is considerable noise in the synthetic borehole data and the various estimates of $T_{y,b}$ and T_b can be quite different with very poor fits of $dT_{y,b}$ to years. I find it disconcerting that by simply including or excluding the reference years in the regression of $dT_{y,b}$ versus years, you can get such different estimates of the degree of warming and these estimates can differ substantially from the estimate provided by T_b .

Again, I don't think that there is anything systemically wrong with the estimates of warming that the authors have produced, especially if the warming trends they observed were all reasonably strong. But if the warming trends were weak, the estimates of decadal warming could be quite uncertain, simply due to the natural error variance in the data. I personally think that a treatment that excludes the reference years from the regression is more defensible. However, it would still be possible to estimate “decadal” changes from this data.

I do not intend to hold this paper up further, over this point. We can disagree about the most appropriate way to define and use the reference temperature for this analysis. As I've noted in previous reviews, I think this is an important paper and considerable care has been taken with the analyses. But it would reassure me if the authors could state that the sort of uncertainty I've describe above is unlikely to have affected their estimates of the regional and global warming trends because of the nature of the underlying data (i.e., the trends were reasonably consistent, monotonic warming).

Other comments to consider

Lines 145-149: This is an awkward sentence. It is very long, chopped up, and the verb is at the end of the sentence. Furthermore, it is not clear to me how the information presented in the sentence actually supports the hypothesis. Start by stating that the "Calculated decadal rates of change in air temperature from ERA Interim data at the borehole locations support this hypothesis." Then explain specifically how the data provide this support. For example, if you mean that the SATs trends are in the same direction and general order of magnitude as the borehole temperatures, then say that.

Lines 165-168: This does not seem to be strong evidence for the difference between the discontinuous and continuous zones.

Lines 168-170: Is the statement relevant to the discontinuous zone, continuous zone, or both. The reference to Figs. e and f suggests both. But that does not make sense. Does this refer specifically to Fig. 4f only for the discontinuous zone?

Line 170: Start a new sentence. "All of these changes provide..."

Line 173-177: And so should I wonder if meaningful correlations can be interpreted from the data?

Line 179: Add changes in "mean annual temperature"

Line 184-186: Okay. But you need to at least briefly say why. Because these effects would have disseminated long ago?

Lines 187-189: It is not clear until here that you are working through alternative explanations for the different trends in ground and air temperatures in the continuous and discontinuous permafrost zones. If that is the intention, then the alternatives the you are comparing should be more clearly identified up front. Perhaps around line 162 you could add a statement that "There are several potential explanations for this difference including...". Then discuss each of these alternatives concluding with your favored explanation.

Lines 195-198: This sentence seems redundant with the next sentence, which I think is a better statement. I suggest that you delete this sentence.

Line 208: For some reason it strikes me as odd to think of C as thawing. Perhaps edit to say "...nor prevent the eventual thawing of permafrost and subsequent decomposition of carbon it contains, which will contribute..."

Lines 283-285: What does this mean?

Line 297: Am I correct that this category would only include boreholes between the minimum of 10 m (line 289) and 12m?

Line 318-319: See my extended comments and Excel spreadsheet about this, above.

Lines 755-756: I understand the intent of the shading between the SO and SE lines; but it is somewhat distracting. I don't think it significantly improves the interpretation of the graph. Furthermore, it is not explained in the legend. Either explain it or remove it.

Lines: 761-762: It would be helpful to provide some indication of whether the median values of each zone are significantly different from zero and from each other. Perhaps use "*" to indicate significant differences from zero and letters (A, B, C...) to indicate differences among zones.

Line 778-779: It would be helpful to indicate or state whether the two medians in the two graphs are significantly different from each other.

How does the slope $\Delta T_{y,b}$ (regression of Eqn 1) relate to ΔT_b (Eqn 2)?

To	-7 oC	Starting temperature(arbitrary)
dT*	0.5 oC	Max annual temperature excursion (arbitrary)
RLB	-45 random lower bound	Set between 0 and -100. More negative than -50 creat
RUB	55 random upper bound	By definition RUB = 100+RLB
Range	100 RUB-RLB	Check. Keep this to 100 units

Randomly generated dataset (9 years)

Year	dTy	Ty	dTy,b (Eqn. 1)
2008	0	-7.0	-0.1
2009	0.29	-6.9	0.1
2010	-0.16	-6.9	0.0
2011	0.29	-6.8	0.1
2012	0.19	-6.7	0.2
2013	-0.43	-6.9	0.0
2014	0.54	-6.6	0.3
2015	0.47	-6.4	0.5
2016	-0.23	-6.5	0.4

This term is common to Eqn 1 and 2

$$\frac{1}{2}(T_{2008}+T_{2009}) \quad -6.9$$

Compare:	All Years	2010-16
Slope dTy,b vs Year	0.062	0.074
Total change (9y)	0.561	0.665
r2 for Total Change	0.724	0.664
dTb (defined by Eqn. 2)	0.465	

tes cooling trend; more positive than -50 creates warming trend.

Hit Fn-F9 to generate a new random combination

Revision Notes

03 November 2018, Biskaborn et al.

Reviewer #3 and answers:

Reviewer #3 (Remarks to the Author):

I appreciate the extensive revisions made by the authors. I've read the rebuttal comments and the revised paper. I conclude that revised paper is much improved. I do have a lingering concern about the analysis done with equations 1 and 2 and have identified a few other comments that think the authors should consider.

Our answers

Dear Dr Bowden

Thank you very much for your careful 3rd review of our manuscript. We assure you that we read your comments very carefully and tested your Excel file fully. Below we explain that there was a misunderstanding caused by poor wording in the last version, and that our analyses were before and are now in line with your model. We assure you and we demonstrate now clearly that there is no major bias.

We are grateful for the additional comments that helped to improve the quality and readability of the manuscript. We hope you agree that we mention your name in the acknowledgements.

Extended comments on Equations 1 and 2

I continue to have some reservation about the uses of equations 1 and 2. Equation 1 provides the departure of the mean borehole temperature from a reference temperature for each year of the record (T_y, b). The regression of these annual values over years provides an estimate of the annual average change in temperature over the period of record. The total change in borehole temperature over the period of record can be determined by solving this regression equation for the first and last years of the record and taking the difference. Equation 2 produces a second estimate of the total difference in

borehole temperature (T_b) by taking the mean difference between the first two years of the record and the last two years of the record. It is noteworthy that the reference temperature used in equation 1 is identical to the “initial” term in equation 2; i.e., it is the mean of the first two years of the record.

Yes, this is correct. Please note that we do not use regression in Eqn 1 and 2 or on data calculated by these equations.

I don't have any particular difficulty with either of these equations and understand why and how the authors use them. I think they are somewhat redundant, but the authors can certainly use both if they wish. I have a somewhat greater concern about a potential instability that could be introduced in the regression of the values generated by equation 1, utilizing the reference temperature as it is defined (the mean of the first two years of the record). I have not been able to articulate this concern in my previous narrative comments and so I have included a simple Excel “thought experiment” that I hope illustrates my point. This file is provided separately.

We fully understand your concern and we apologize for not having stated this clear enough in the manuscript or in our responses in the peer-review process. We assure you that we do not apply regression „utilizing the reference temperature as it is defined (the mean of the first two years of the record).“ We only use means of multiple years to base the departures from year to year on a more stable reference period, as established in climate change calculations. But we do not use regression on departures, we use regression on the original T data.

In this thought experiment I have assumed arbitrarily that the initial permafrost temperature is -70°C (cell B3). I also assumed that there is some error in each annual measurement that is a combination of real and stochastic events. I limited this error to a maximum of 0.50°C (cell B4) in any given year. The choice of the actual values for these two variables is not particularly important. I then introduced a random factor that has a dynamic range of 100. The maximum allowable lower bound on this dynamic range is -100 and the maximum allowable upper bound is $+100$, but the dynamic range itself can only be 100 units. Thus, this factor is perfectly “centered” if the random lower bound (RLB, cell B5) is -50 and random upper bound (RUB, cell B6) is $+50$. In this case the Excel random number generator will on average produce as many negative numbers (between 0 and -50) as it does positive number (between 0 and $+50$). If the boundaries are skewed upward so that $\text{RLB} = -25$ then the RUB must be $+75$ (to preserve a dynamic range of 100). In this case the Excel random number generator will produce more positive numbers (between 0 and $+75$) than it does negative numbers (between 0 and -25). In use, each random number that is generated (each year) is divided by 100 to create a fraction. That fraction is multiplied by the 0.50°C maximum temperature excursion and then added to the previous

year's borehole temperature. This is done for each year (cells A11 to A19) to create a synthetic borehole record (cells C11 to C19) from the random annual temperature excursions (cells B11 to B19). If the dynamic range is "centered" then the temperature record is simply random variation around the mean value of -7oC. But if the dynamic range is skewed upward, it generates a "warming" trend and if it is skewed downward it generates a "cooling" trend. The magnitude of the skew determines the "strength" of the trend. A new realization of this model can be created by clicking the Function-F9 keys together. This will re-randomize the model and quickly allows the user to see numerous iterations of the same set of conditions.

We are very grateful for this impressive work, which is obviously able to model permafrost temperature error estimation in an advanced way using Excel functions. There is a great learning from this for various scientific purposes. However, there is one misunderstanding - which is probably our fault, because we did not separate between the explanations of Eqn1 and Eqn2 on the one hand and the regression analysis on the other hand while describing the methods. We stated in the same paragraph where we explained Eqn1 and Eqn2 the following explanation to the applied regression: „To calculate the rate of temperature change per year we applied linear regression on $\bar{T}_{y,b}$ for all $y \in i$ using the linear model function (lm) in the R environment“. However, the regression performed in the Excel thought experiment is performed on the results of Eqn1 which would be $\Delta\bar{T}_{y,b}$ including reference period values. This is the misunderstanding. Our sentence referred to primary data $\bar{T}_{y,b}$ ($\neq \Delta\bar{T}_{y,b}$). We apologize very much because we believe that it is our fault not having separated clear enough between these different approaches. In addition, we admit that it is more correct to state "regression used on \bar{T}_b " (without indexed y), because we use all available primary temperature data in a borehole from 2007-2016. Both imprecise wordings on our side probably led to the misunderstanding. Your conclusion based on the mathematical experiment provided in the additional file is an independent proof, that the methodological approach in the manuscript is valid. In fact, your personal recommendation is similar to what was conceived over years designing this study involving the knowledge of numerous experiences permafrost researchers (with the exception that we can use the full time series because we don't need to use departures for the calculation). To state this as clear as possible in the manuscript, we now explained the regression method in a new Eqn 3 and rephrased:

We define a set $i = \{2007, \dots, 2016\}$ to identify the years. To identify the boreholes b we use the GTN-P Database ID. Continuous (full-year) records started at a large number of borehole sites in 2008, the second year of the 4th International Polar Year (IPY). To base the reference period on the annual departure calculation on the largest possible number of boreholes we exclude 2007 and estimated the annual differences in \bar{T} in year $y \in i$ and borehole b as

$$\Delta \bar{T}_{y,b} = \bar{T}_{y,b} - 1/2 (\bar{T}_{2008,b} + \bar{T}_{2009,b})$$

The last term on the right-hand side of equation (1) serves as our mean value for the reference period. We compare this reference period to the latest available mean value period and calculate $\Delta \bar{T}_b$ to rank total temperature differences among boreholes.

$$\Delta \bar{T}_b = 1/2 (\bar{T}_{2015,b} + \bar{T}_{2016,b}) - 1/2 (\bar{T}_{2008,b} + \bar{T}_{2009,b})$$

Equations (1) and (2) require data to be available in each of the observation years. To calculate the rate of temperature change per decade we follow a third approach using the primary mean annual ground temperature data set \bar{T}_b for all available years in i and perform linear regression, according to the following attribution of our data in the regression equation:

$$\bar{T}_b^{reg} = a_b + c_b x$$

where \bar{T}_b^{reg} is the regression estimate of \bar{T}_b , a_b is the vertical intercept (the starting temperature in a borehole), c_b is the slope of the regression line, and x is the range of years involved.

*The Excel spreadsheet calculates $dT_{y,b}$ in cells D11 to D19 (highlighted in yellow) and also the reference temperature ($1/2 * [T_{2008} + T_{2009}]$) in cell H10. This data is then used to generate the data in the block of cells from E12 to I16 (also highlighted in yellow). This block calculates the slope of $T_{y,b}$ by year, the total change expected over all years (9 in this case), the r^2 for this regression, the value of dT_b from Equation 2. It does this for two different treatments of the data. The first treatment (cells H13 to H15) is the treatment as described in the paper, in which the reference years are defined as in cell H10 and all years in the record are assessed against that reference. The second treatment of the data (cells I13 to I15) excludes the data for the reference years from the regression, so as not to “double count” their influence on the regression. This is the situation I attempted to describe in my previous review. The value of dT_b in the merged cell at H/I16 is the same for both treatments. But it can be compared to the alternative forms of the same metric (the total change over 9 years, the “decadal” change) for the two different treatments of the data (cells HH4 and I14.)*

You are exactly right with your assumptions and we fully tried out and understood your thought experiment. We especially acknowledge the intelligent way of proving, that utilization of Eqn1 in regression analysis would result in a skewed temperature transfer. This is correct. However, in your Excel sheet, the value „Total change (9y)“ is based on the results of our Eqn1 and compares this with an alternative approach losing out the reference period. You demonstrate a significant bias in the first case. As you stated, the best way to avoid this bias is to exclude the reference period (double count of years) from this equation. This is correct - and this is exactly what our calculation does. It was perhaps our mistake to not state clear enough our method. We stated this now unambiguously in the manuscript by adding another equation. Please see our explanation above.

My conclusion from playing with this model is that if the warming trend is strong (i.e., the skew of the dynamic range is say -5 to 95), then repeated re-randomizations of the model typically create a well-behaved, upward, linear trend and the various estimates of $T_{y,b}$ and T_b are reasonably similar with high r^2 values for the regression of $dT_{y,b}$ versus years.

However, if the warming trend is weak (say, a $RLB = -45$ and $RUB = +55$) then there is considerable noise in the synthetic borehole data and the various estimates of $T_{y,b}$ and T_b can be quite different with very poor fits of $dT_{y,b}$ to years. I find it disconcerting that by simply including or excluding the reference years in the regression of $dT_{y,b}$ versus years, you can get such different estimates of the degree of warming and these estimates can differ substantially from the estimate provided by T_b .

Again, I don't think that there is anything systemically wrong with the estimates of warming that the authors have produced, especially if the warming trends they observed were all reasonably strong. But if the warming trends were weak, the estimates of decadal warming could be quite uncertain, simply due to the natural error variance in the data. I personally think that a treatment that excludes the reference years from the regression is more defensible. However, it would still be possible to estimate "decadal" changes from this data.

Thank you very much again for this remarkable model. Our calculation is in accord with your personal recommendation. Please accept our apology for not having this stated clear enough in the methods, which we did in the new version. In addition, there is one more proof that the presented results of permafrost T change is indeed reliable: The error limitation you applied was 0.5°C . In permafrost, as we show in the accuracy method section, this limit can be set to 0.1°C and thus the overall error is much lower.

I do not intend to hold this paper up further, over this point. We can disagree about the most appropriate way to define and use the reference temperature for this analysis. As I've noted in previous reviews, I think this is an important paper and considerable care has been taken with the analyses. But it would reassure me if the authors could state that the sort of uncertainty I've describe above is unlikely to have affected their estimates of the regional and global warming trends because of the nature of the underlying data (i.e., the trends were reasonably consistent, monotonic warming).

Thank you very much for your careful revision. We have shown, that the uncertainty you described does not affect our estimates. The temperature trends observed are reasonably consistent and the observed mean warming rates exceed the error limit. We fully understand and verify the integrity of your concerns shown precisely in a thought experiment. We apologize again for not having separated clear enough between the methods we applied in the manuscript text.

A very good and independent proof on the integrity of the methods is that you recommended to apply a calculation concept which is matching the calculation performed (but not stated clear enough before). Therefore, it only was necessary to modify the text in the method section but not the calculations.

Other comments to consider

Lines 145-149: This is an awkward sentence. It is very long, chopped up, and the verb is at the end of the sentence. Furthermore, it is not clear to me how the information presented in the sentence actually supports the hypothesis. Start by stating that the "Calculated decadal rates of change in air temperature from ERA Interim data at the borehole locations support this hypothesis." Then explain specifically how the data provide this support. For example, if you mean that the SATs trends are in the same direction and general order of magnitude as the borehole temperatures, then say that.

Yes, we agree and revised in two sentences:

"Mean surface air temperature changes calculated from ERA Interim data at the borehole locations (Fig. 5b) are similar to those for permafrost temperature with respect to direction and order of magnitude. The decadal change rates of air temperature were estimated to 0.86 ± 0.84 °C per reference decade in the Arctic continuous permafrost zone, 0.63 ± 0.91 °C $\text{dec}_{\text{Ref}}^{-1}$ in the Arctic discontinuous permafrost zone, and 0.1 ± 0.50 °C $\text{dec}_{\text{Ref}}^{-1}$ in mountain permafrost."

Lines 165-168: This does not seem to be strong evidence for the difference between the discontinuous and continuous zones.

Differences in the snow cover have been proven to be the dominant driving factor of the formation of permafrost in transitional regions between continuous and discontinuous zone (see references cited in the manuscript, i.e. TSP snapshot papers 2010, references 2-7). We added one more reference to a study proving that a difference of 10 days in the timing of snow already causes permafrost warming on large scale in Alaska. We also added a referenced sentence on the increase in height and density of shrubs leading to trap wind drifting snow. We confirm that snow is one of multiple driving factors for permafrost T change differences in these two zones, next to air T changes and the latent heat effect.

Lines 168-170: Is the statement relevant to the discontinuous zone, continuous zone, or both. The reference to Figs. e and f suggests both. But that does not make sense. Does this refer specifically to Fig. 4f only for the discontinuous zone?

We agree. It is referred to the discontinuous zone referred to in the sentence before and to Fig.4f. We revised the sentence to: "Compared to 2007–2009 the

snow cover in 2014–2016 in the discontinuous zone started to form 13.7 days earlier, reached its maximum insulation effect 37.7 days earlier, and disappeared 9.3 days earlier (Fig. 4f)."

Line 170: Start a new sentence. "All of these changes provide..."

We agree and revised accordingly.

Line 173-177: And so should I wonder if meaningful correlations can be interpreted from the data?

We agree. Our results indicate the relationships demonstrated in the manuscript and it is not necessary to mention that system Earth is complex. We assume that this is common knowledge and deleted this paragraph.

Line 179: Add changes in "mean annual temperature"

We agree and revised accordingly.

Line 184-186: Okay. But you need to at least briefly say why. Because these effects would have disseminated long ago?

We agreed among authors to remove this sentence because mentioning of the little ice age could confuse reader (our study does not concern the last centuries).

Lines 187-189: It is not clear until here that you are working through alternative explanations for the different trends in ground and air temperatures in the continuous and discontinuous permafrost zones. If that is the intention, then the alternatives the you are comparing should be more clearly identified up front. Perhaps around line 162 you could add a statement that "There are several potential explanations for this difference including...". Then discuss each of these alternatives concluding with your favored explanation.

We agree and added one sentence at the suggested position before discussion the different driving factors. "We found that snow dynamics, the time lag between air and ground temperature, and the latent heat effect serve as concurrent explanations for this phenomenon."

Lines 195-198: This sentence seems redundant with the next sentence, which I think is a better statement. I suggest that you delete this sentence.

We agree. We deleted parts of the paragraph and modified to: “The warming of permafrost observed since IPY continues the trends documented prior to IPY⁴⁰. Our global analysis suggests that the future increases in air temperature projected under current climate scenarios⁹ will result in continued permafrost warming”.

Line 208: For some reason it strikes me as odd to think of C as thawing. Perhaps edit to say “...nor prevent the eventual thawing of permafrost and subsequent decomposition of carbon it contains, which will contribute...”

We agree and revised to: “...nor prevent the eventual thawing of permafrost. This could have wide implications in terms of permafrost degradation and release of greenhouse gases from decomposition of organic matter”.

Lines 283-285: What does this mean?

We added an explanation after this text passage: “Thus, the data distribution represents an approximation to Z^* which minimizes the potential bias caused by seasonal fluctuations.”

Line 297: Am I correct that this category would only include boreholes between the minimum of 10 m (line 289) and 12m?

We generally used the depth at or nearest to Z^* , which can be 20m or deeper. We revised to: “Statistically indifferent temperature trends of the remaining shallow (≤ 12 m) and deeper (>12 m, max. 40 m) boreholes in the utilized data set confirm that the observed depths near Z^* (Fig. 6b) provide a representative sample tracking climate variability coherently”.

Line 318-319: See my extended comments and Excel spreadsheet about this, above.

Please see our answer above.

Lines 755-756: I understand the intent of the shading between the SO and SE lines; but it is somewhat distracting. I don't think it significantly improves the interpretation of the graph. Furthermore, it is not explained in the legend. Either explain it or remove it.

Thank you. We agree and removed the shading.

Lines: 761-762: It would be helpful to provide some indication of whether the median values of each zone are significantly different from zero and from each other. Perhaps use "" to indicate significant differences from zero and letters (A, B, C...) to indicate differences among zones.*

We agree and added significances for the differences to 0 (Wilcoxon Signed-Rank test) and the differences among zones (Kruskal-Wallis test) in the graph and explained the new graph elements in the captions. We consequently added and explained the insignificant trend of Antarctic permafrost temperature change in the main text (line 110-112).

Line 778-779: It would be helpful to indicate or state whether the two medians in the two graphs are significantly different from each other.

We agree and added significances for the differences to 0 (Wilcoxon Signed-Rank test) and the differences among zones (Kruskal-Wallis test) in the graph and explained the new graph elements in the captions. We also added "Statistically indifferent temperature trends..." in the method section (lines 281-284) to validate the chosen depths range.

Thank you very much. Your effort on reviewing our manuscript helped very much to assure the quality of this study.

We additionally fine-tuned the English in the entire manuscript again and added the grant numbers of the projects that performed permafrost temperature monitoring and data analysis in the acknowledgement chapter.

With kind regards,
Boris Biskaborn et al.